# TRIM28-dependent SUMOylation protects the adult ovary from activation of the testicular pathway

Moïra Rossitto[1,9], Stephanie Déjardin [1], Chris M. Rands[2,10], Stephanie Le Gras[3,10], Roberta Migale[4], Mahmoud-Reza Rafiee [4], Yasmine Neirijnck [2], Alain Pruvost [5], Anvi Laetitia Nguyen[5], Guillaume Bossis [6], Florence Cammas [7], Lionel Le Gallic[1], Dagmar Wilhelm [8], Robin Lovell-Badge [4], Brigitte Boizet-Bonhoure[1], Serge Nef [2] & Francis Poulat [1] ✉

Gonadal sexual fate in mammals is determined during embryonic development and must be actively maintained in adulthood. In the mouse ovary, oestrogen receptors and FOXL2 protect ovarian granulosa cells from trans-differentiation into Sertoli cells, their testicular counterpart. However, the mechanism underlying their protective effect is unknown. Here, we show that TRIM28 is required to prevent female-to-male sex reversal of the mouse ovary after birth. We found that upon loss of *Trim28*, ovarian granulosa cells trans-differentiate to Sertoli cells through an intermediate cell type, different from gonadal embryonic progenitors. TRIM28 is recruited on chromatin in the proximity of FOXL2 to maintain the ovarian pathway and to repress testicular-specific genes. The role of TRIM28 in ovarian maintenance depends on its E3-SUMO ligase activity that regulates the sex-specific SUMOylation profile of ovarian-specific genes. Our study identifies TRIM28 as a key factor in protecting the adult ovary from the testicular pathway.

For long time, it was thought that in mammals, adult gonadal sex assignment was determined and fixed during embryonic development. Any perturbation during this period leads to various disorders of sexual development. However, some teleost fish species display sequential hermaphroditism: gonadal sex is not definitively established in adulthood, and social stimuli can re-assign gonads to the opposite sex (for review see[1]). Moreover, postnatal sex reversal has been observed in several mouse models: ovarian masculinisation upon

deletion of oestrogen receptor 1 and 2 (*Esr1-2*)[2] or of *Cyp19a1*[3], as well as after postnatal conditional knock-out (cKO) of *FoxL2*[4] and ectopic ovarian expression of *Dmrt1*[5]. In these cases, the initial cellular event is ovarian-to-testicular transdifferentiation of the supporting cell lineage (granulosa cells to Sertoli cells). Conversely, deletion of *Dmrt1* in postnatal testes[6] or of both *Sox8* and *Sox9*[7] induces Sertoli-to-granulosa cell transdifferentiation. These results indicate that granulosa and Sertoli cells retain the ability to transdifferentiate into the

[1]Institute of Human Genetics, CNRS UMR9002 University of Montpellier, 34396 Montpellier, France. [2]Department of Genetic Medicine and Development, Faculty of Medicine, University of Geneva CMU, lab E09.2750.B 1, rue Michel-Servet CH 1211 Geneva 4, Geneva, Switzerland. [3]GenomEast platform, IGBMC, 1, rue Laurent Fries, 67404 ILLKIRCH Cedex Illkirch-Graffenstaden, France. [4]The Francis Crick Institute, 1 Midland Road, London NW1 2 1AT, UK. [5]Université Paris Saclay, CEA, INRAE, Département Médicaments et Technologies pour la Santé (DMTS), SPI, 91191 Gif-sur-Yvette, France. [6]Institut de Génétique Moléculaire de Montpellier (IGMM), University of Montpellier, CNRS, Montpellier, France. [7]Institut de Recherche en Cancérologie de Montpellier, IRCM, INSERM U1194, Université de Montpellier, Institut régional du Cancer de Montpellier, Montpellier F-34298, France. [8]Department of Anatomy and Physiology, University of Melbourne, Parkville, VIC 3010, Australia. [9]Present address: Univ. Bordeaux, INRAE, Bordeaux INP, NutriNeuro, UMR 1286, F-33000 Bordeaux, France. [10]These authors contributed equally: Chris M Rands, Stephanie Le Gras. ✉e-mail: francis.poulat@igh.cnrs.fr

opposite sexual fate, and that constant repression of the alternative fate in adult life is required to maintain their cell fate identity and function. However, there is only limited information on the epigenetic and transcriptional programmes implicated in cell fate reprogramming of the supporting lineage.

We previously showed that the epigenetic regulator TRIM28 is a partner of SOX9 in mouse fetal Sertoli cells[8]. TRIM28 is a versatile nuclear scaffold protein that coordinates the assembly of protein complexes containing different chromatin remodelling factors. It can be recruited on chromatin upon interaction with DNA-binding proteins, such as KRAB-ZNF family members[9–11], or with transcription factors[12–14]. TRIM28 was originally associated with transcriptional repression[9] and heterochromatin formation[15,16]; however, many evidences show that it also positively regulates gene expression[12,13,14,17] and controls transcriptional pausing[18,19]. Despite its interaction with SOX9, cKO of *Trim28* in Sertoli cells results in adult males with hypoplastic testes and spermatogenesis defects, but no sex reversal[20]. This suggests that in Sertoli cells, TRIM28 is required to control spermatogenesis, but not for the maintenance of the somatic cell component of the testis.

In this work, to understand its role in ovarian physiology, we generated a cKO of *Trim28* in the somatic compartment of the developing mouse ovary. We observed sex reversal in adult ovaries where the follicular structure progressively reorganised in pseudo-tubules with Sertoli-like cells. We then combined mouse genetic with transcriptomic and genomic approaches to determine the molecular action of TRIM28 and its interplay with FOXL2 in adult ovaries. Our data show that TRIM28 maintains the adult ovarian phenotypes through its SUMO-E3-ligase activity that controls the granulosa cells programme and represses the Sertoli cell pathway.

## Results

### Deletion of *Trim28* induces masculinisation of adult ovary

Double immunostaining of XX gonads at 13.5 days post-coitum (dpc) showed that TRIM28 is co-expressed with FOXL2 in ovarian pre-granulosa cells, (Supplementary Fig. 1). To study its role in this crucial ovarian lineage, we generated a mouse line in which *Trim28* can be conditionally deleted using the *Nr5a1:Cre*[21,22] transgenic line (*Trim28^flox/flox^*; *Nr5a1:Cre* referred as *Trim28^cKO^* or cKO in the text/figures). In 13.5 dpc cKO ovaries, nuclear TRIM28 signal was strongly decreased in FOXL2-positive pre-granulosa cells, whereas it was still present at heterochromatin foci, and was nearly disappeared at E18.8 (Supplementary Fig. 1). At birth, XX cKO mice displayed normal external female genitalia, without any obvious ovarian structure abnormality at 3 days post-partum (dpp)(Supplementary Fig. 2). In FOXL2-positive immature granulosa cells, we did not detect any signal for TRIM28 and SOX8/SOX9, two Sertoli cell markers (Supplementary Fig. 2). Unlike granulosa cells that looked normal at this stage, oocytes were larger, suggesting an early and indirect effect of TRIM28 absence on oogenesis. This suggests that TRIM28 is not required for fetal ovary differentiation. However, as TRIM28 is still expressed in pre-granulosa cells at 13.5 dpc, a potential role in the primary ovarian determination that occurs at ~11.5 dpc cannot be excluded.

In several follicles of 20 dpp *Trim28^cKO^* ovaries, SOX8 was expressed in groups of cells that stopped expressing FOXL2 (Fig. 1a). Some of these cells are co-expressing FOXL2 and SOX8, suggesting a transdifferentiation event. Double immunostaining showed that some SOX8-positive cells also expressed SOX9, suggesting that SOX8 expression precedes SOX9, unlike what observed in mouse embryonic testes[23]. As SOX8 and SOX9 are Sertoli cell markers, this suggests that fetal deletion of *Trim28* in pre-granulosa cells might induce their reprogramming towards Sertoli cells after birth, as described for *Foxl2* deletion[4] and oestrogen receptor double knock-out[2].

In 8-week-old *Trim28^cKO^* mice, ovarian organisation was profoundly changed. Medullar follicles had almost completely lost FOXL2 expression, expressed SOX8 and SOX9, and were reorganised into pseudo-tubular structures, indicative of a process of testis cord formation (Supplementary Fig. 3a, d, g). We never detected any cell that expressed both SOX8 (Supplementary Fig. 3b) or SOX9 (Supplementary Fig. 3e) and FOXL2, but many cells that expressed both SOX proteins (Supplementary Fig. 3h). Their distribution suggested (like in 20 dpp *Trim28^cKO^* ovaries) that SOX8 might precede SOX9. Conversely, the cortical region presented a less advanced phenotype: as observed in 20 dpp *Trim28^cKO^* ovaries, follicles were still organised, but remodelling had started with groups of cells that stopped expressing FOXL2 and expressed SOX8 and/or SOX9 (Supplementary Fig. 3c, f and i). These results show that in *Trim28^cKO^* ovaries, the granulosa-to-Sertoli cell transdifferentiation starts in follicles located in the medulla and then spread to the cortical regions.

In parallel, using the Terminal deoxynucleotidyl transferase dUTP nick end labelling (TUNEL) assay, we did not observe any significant increase in apoptosis in 20 dpp and 8-week-old *Trim28^cKO^* ovaries (Supplementary Fig. 4), as previously described for the cKO of *Foxl2*[4]. This excluded the replacement by neo-formed Sertoli cells of granulosa cells eliminated by widespread apoptosis.

In 4-month-old *Trim28^cKO^* females, the transdifferentiation of granulosa cells into Sertoli cells was complete: FOXL2 expression has disappeared, and follicles were completely remodelled into tubular structures with cells that expressed the Sertoli cell markers SOX8, SOX9 and DMRT1 (Fig. 1b). Histological analysis confirmed the progressive reorganisation of ovarian follicles into tubular structures and the transdifferentiation of granulosa cells into cells with a Sertoli cell morphology (Supplementary Fig. 5). This reorganisation was undetectable in 4-week-old *Trim28^cKO^* ovaries but was clearly visible in the medulla at 8 weeks and was completed at 17 weeks. Germ cells (oocytes) were relatively normal in ovaries with a preserved follicular structure but started to degenerate during transdifferentiation. In 8-week-old ovaries in which the medullar part was reorganised into pseudo-tubules, oocytes had disappeared or were degenerating (Supplementary Fig. 5), and in 17-week-old ovaries they had disappeared.

A recent study showed that *Trim28* hemizygosity affects spermatogonial stem cells and induces testis degeneration[24]. However, we did not observe any change in FOXL2 immunostaining in ovaries from wild-type and heterozygous *Trim28^cKO^* mice at the different stages we analysed (Supplementary Fig. 6a). Similarly, we did not detect any expression change of the three Sertoli markers *Sox8*, *Sox9* and *Dmrt1* in heterozygous 3-month-old ovaries (Supplementary Fig. 6b). Therefore, the loss of a single *Trim28* allele does not cause transdifferentiation of granulosa cells.

We next examined the temporal expression of several genes with roles in testicular and ovarian sex-determination in 0.5- (15 dpp), 2 and 4-month-old ovaries. Reverse transcription-quantitative real-time polymerase chain reaction (RT-qPCR) analysis revealed that in *Trim28^cKO^* ovaries, the mRNA level of most ovarian-specific genes was decreased, with the exception of *Rspo1* (Fig. 1c, panel Ovarian genes). Conversely, testicular-specific genes were progressively upregulated (Fig. 1c, panel Testicular genes), confirming the histology and immunofluorescence observations. The expression level of some ovarian (*Foxl2*, *Esr2*, *Cyp19a1*, and *Rspo1*) and testicular genes (*Sox8* and *Dhh*) was already modified soon after birth (15 dpp), before changes in *Sox9* and *Dmrt1* and before the detection of histological defects (Supplementary Fig. 5).

Bulk RNA-seq experiments using 7-month-old *Trim28^cKO^* ovaries (Data S1), in which transdifferentiation was completed, showed that 1669 genes were significantly downregulated in the absence of *Trim28*, among which 71% are normally expressed in adult granulosa cells[25], including genes involved in ovarian determination (Fig. 1d, right).

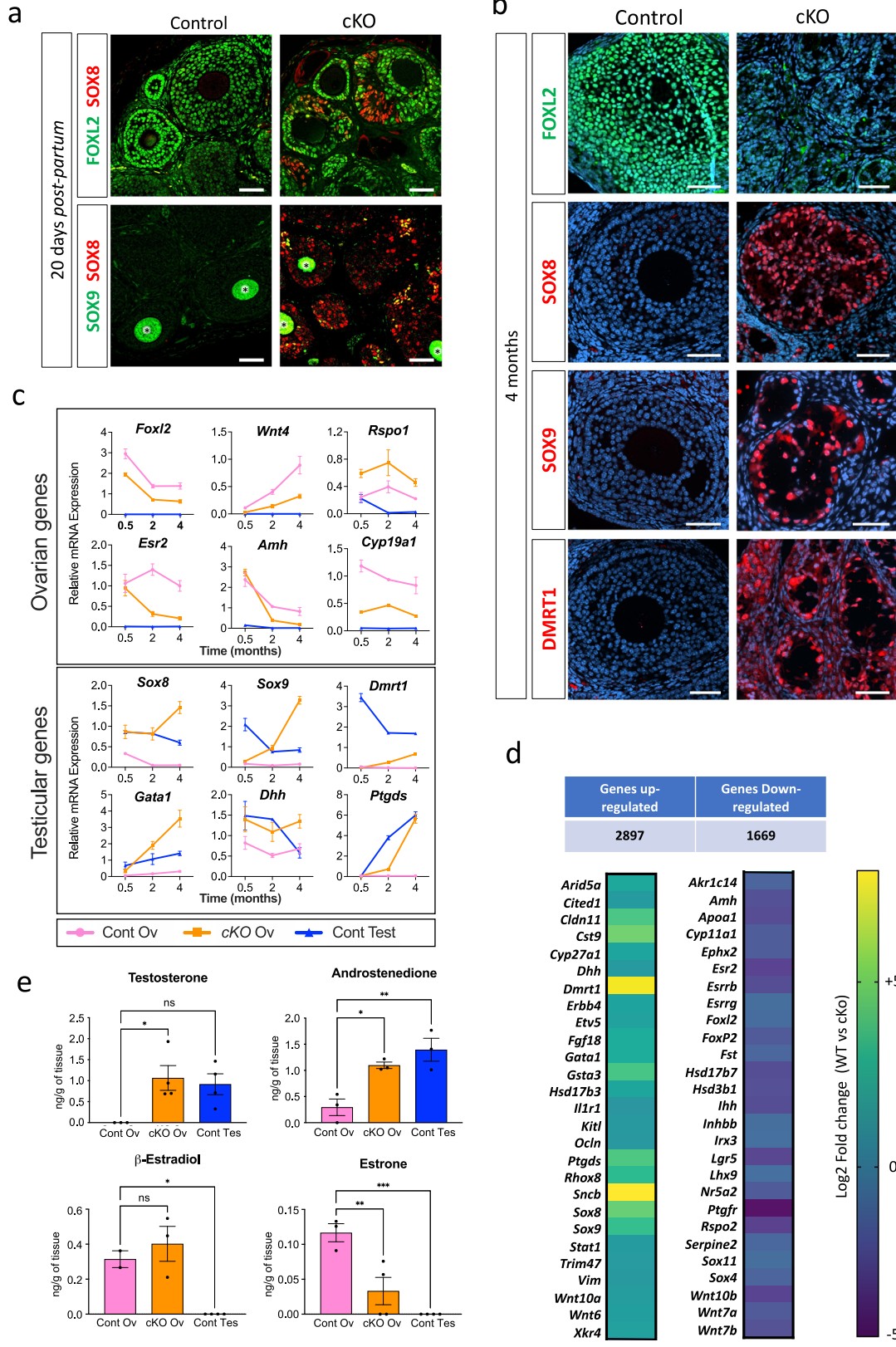

Repression of the granulosa cell transcriptome was accompanied by upregulation of 2897 genes that included typical Sertoli and Leydig cell markers (Fig. 1d, left), showing that *Trim28* cKO induces the ovarian transcriptome masculinisation. We concluded that *Trim28* deletion in

fetal pre-granulosa cells induces the postnatal remodelling of the ovarian transcriptome, leading to its masculinisation. Moreover, we observed an important deposition of extracellular matrix around pseudo-tubules (Supplementary Fig. 5) and the upregulation of several

**Fig. 1 | *Trim28* loss in granulosa cells induces masculinisation of the adult ovary. a** compared with control ovaries, in granulosa cells of 20 dpp *Trim28^cKO* ovaries, FOXL2 expression is progressively lost and SOX8 (Sertoli cell marker) starts to be expressed. An overlap of both stainings is also visible, showing that some cells are co-expressing FOXL2 and SOX8. Among the SOX8-positive cells, few express also SOX9, suggesting that SOX8 may precede SOX9. Green staining of oocytes (*) is a non-specific antibody artefact of early folliculogenesis[105]. Scale bar: 50 μm. **b** in 4-month-old *Trim28^cKO* ovaries, transdifferentiation to Sertoli cells is complete. Compared with control ovaries, in *Trim28^cKO* ovaries FOXL2 signal has almost disappeared, and follicles are reorganised in pseudo-tubules that express the Sertoli markers SOX8, SOX9, and DMRT1. Protein (green or red) are merged with DNA stain (blue). Scale bar: 50 μm. **c** RT-qPCR analysis of the temporal (in months) gene expression variations in control ovaries (Cont Ov), *Trim28^cKO* ovaries (cKO Ov), and control testes (Cont Test). In *Trim28^cKO* ovaries, typical ovarian genes are progressively downregulated, but for *Rspo1*, and testis genes are upregulated. Bars are the mean ± SEM. Details of the statistical analysis are provided in Source data file. Source data are provided as a Source Data file. Bars are the mean ± SEM. For 0.5, 2 and 4 months: control ovaries $n = 5, 4, 4$ animals (gonad pairs) respectively; cKO ovaries

$n = 5$ animals, control testes: $n = 3$ animals. Details of the statistical analysis are provided in Source data file. **d** Heatmap of the RNA-seq analysis of 7-month-old ovaries (see Data S1) showing that 2896 and 1669 genes are up- and downregulated, respectively, in *Trim28^cKO* compared with control ovaries. Normalised expression values are expressed as Log2 fold-change (Control *vs* cKO), from −5 (deep violet) to +8 (yellow). Source data are provided as a Source Data file. **e** *Trim28* cKO induces the masculinisation of the ovarian steroid profile. Steroids were extracted from 7-month-old control (Cont Ov) and *Trim28^cKO* (cKO Ov) ovaries, and control testes (Cont Test) and quantified (ng/g of tissue) by mass spectroscopy. Data are the mean ± SEM. For testosterone $n = 4$ animals (gonad pairs). For Androstenedione $n = 3$. For β-estradiol $n = 3,3, 4$ animals for Cont Ov, cKO Ov and Cont Test respectively. For estrone $n = 3,4, 4$ animals for Cont Ov, cKO Ov and Cont Test respectively. *P* value for Testosterone *:0.0324; for androstenedione * and**: 0.2013 and 0.005 respectively; for β-estradiol *: 0.0215; for estrone ** and ***: 0.058 and 0.0008 respectively. (One-way ANOVA with Dunnett's multiple comparisons test). Source data are provided as a Source Data file. Details of the statistical analysis are provided in Source data file. For the immunofluorescences, at least three independent biological replicates were analysed, and the images presented are representative of all replicates.

genes that encode components of the testicular basal lamina: *Col4a3*, *Col9a3*, *Col13a1*, *Col28a1*, and *Lamc2* (encoding laminin gamma 2) (Data S1).

As several genes involved in steroidogenesis displayed a modified profile (Fig. 1c, Supplementary Fig. 7a, b for temporal analysis, and RNA-seq data, respectively), we used mass spectroscopy to quantify the production of major steroid hormones in control and *Trim28^cKO* ovaries and control testes from 7-month-old animals (Fig. 1e). Androgen levels (testosterone and androstenedione) in *Trim28^cKO* ovaries and control testes were similar. Among the oestrogens produced in *Trim28^cKO* ovaries, estrone was strongly reduced, whereas 17β-estradiol levels were comparable to those in control ovaries. This can be explained by the persistent expression of *Cyp19a1* (the gene encoding the aromatase that catalyses 17β-estradiol production) in *Trim28^cKO* ovaries (Fig. 1c) and by the modified expression of genes encoding hydroxysteroid dehydrogenases (HSD) (Supplementary Fig. 7a, b). Overall, our results indicate that fetal *Trim28* deletion induces the masculinisation of the steroid production profile in adult ovaries.

## Analysis of the granulosa-to-sertoli transdifferentiation

To better describe the transdifferentiation process, we performed single-cell RNA sequencing (scRNA-seq) to compare the transcriptomic atlas of gonadal cell types in *Trim28^cKO* ovaries, control ovaries, and control testes. We analysed 8-week-old gonads because our data (Supplementary Fig. 3) indicated that at this stage, *Trim28^cKO* ovaries contain a mixed population of Sertoli-like cells and apparently normal granulosa cells. Using the 10X Genomics Single-Cell Gene Expression system, we analysed 7292 cells from *Trim28^cKO* ovaries, 7051 from control ovaries, and 42,720 from control testes (total = 57,063 cells). A larger number of testis cells was required to sample an equivalent number of testicular somatic cells alongside the abundant spermatogenic cells. We catalogued the different cell populations present in all samples (Supplementary Fig. 8a) based on the expression of known markers (Supplementary Fig. 8b). We confirmed the substantial decrease of *Trim28* expression in *Trim28^cKO* ovarian cells (Supplementary Fig. 9). In control gonads, we detected the expected cell types, including supporting (granulosa/Sertoli), steroidogenic (theca/Leydig), stroma, spermatogenic, endothelial, immune and blood cells (Supplementary Fig. 8), consistent with previous single-cell transcriptomic studies of adult mouse/human testis/ovaries[26–28]. We then focused on the supporting cell lineages. We identified 3106 supporting cells that expressed granulosa and/or Sertoli cell markers ($n = 1112$ in *Trim28^cKO* ovaries, $n = 1446$ in control ovaries, and $n = 548$ in control testes) (Fig. 2a). In *Trim28^cKO* ovaries, transcriptional profiles were asynchronous, some supporting cells were grouped with control granulosa cells and expressed *Esr2*, *Amh*, *Foxl2*, *Wnt4*, *Hsd17b1*, and *Nr5a2*, indicating that they still had a granulosa-like transcriptome

(Fig. 2b). However, we also observed a gradient of gene expression from granulosa-to Sertoli cells via some intermediate *Trim28^cKO* ovarian supporting cells (Fig. 2a) that expressed some Sertoli markers at various levels and at different stages of transdifferentiation.

For example, *Cldn11* and *Ptgds* were expressed earlier during transdifferentiation and in more cells, compared with *Gata1*, *Dmrt1*, *Sox9* and *Sox8* (Fig. 2c).

We then asked whether these intermediate cells resembled embryonic XX or XY supporting cell progenitors[29] that dedifferentiated from granulosa cells before differentiating into the Sertoli lineage. We aligned all single cells along a pseudo-time (Fig. 2d, e, Supplementary Fig. 10)[30], and divided them into three clusters based on their transcriptional profiles (Fig. 2a, right). This allowed us to identify genes that were upregulated in the granulosa, intermediate, and Sertoli cell populations (Fig. 2d, Data S3). Analysis of the mean expression of 1,743 supporting progenitor cell markers[29] showed that they were weakly expressed in intermediate cells (Fig. 2e). This indicated that this population was distinct from embryonic progenitors. Gene Ontology enrichment analysis of the genes expressed in the intermediate population gave only general terms, such as "response to stimulus", "cell death", and "cell differentiation" (Data S4). Overall, the scRNA-seq analysis showed that in adult ovaries, *Trim28* cKO leads to transdifferentiation of the supporting lineage from the granulosa-to the Sertoli cell fate. Moreover, granulosa cells do not transdifferentiate into Sertoli cells by returning to an embryonic progenitor state, but via a different and novel cell intermediate (Fig. 2f).

## TRIM28 acts in concert with FOXL2 on chromatin

As the *Trim28^cKO* phenotype was similar to that of mice after *Foxl2* deletion in adult ovarian follicles[4], we asked whether these two proteins co-regulated common target genes in the ovary. Immunofluorescence analysis confirmed that TRIM28 and FOXL2 were strongly co-expressed in the nucleus of adult control follicular granulosa cells and to a lesser extent in theca stromal cells. Both were almost undetectable in *Trim28^cKO* ovaries (Fig. 3a). Next, we performed TRIM28 and FOXL2 chromatin immunoprecipitation (ChIP) followed by next-generation sequencing (ChIP-seq) in control ovaries to gain a global view of TRIM28 and FOXL2 colocalization genome-wide. A comparison of the heatmaps of their co-binding to chromatin (Fig. 3b) showed that in ovaries, FOXL2 ChIP-seq reads strongly mapped to regions occupied by TRIM28 (Fig. 3b, blue panel). Similarly, TRIM28 ChIP-seq reads strongly mapped to FOXL2 peaks (Fig. 3b, red panel). Analysis of the overlap between TRIM28 and FOXL2 peaks confirmed that these proteins shared common genomic targets (62 and 55% respectively, Fig. 3b Venn diagram). TRIM28 and FOXL2 bound to overlapping regions of genes that have a central role in ovarian determination, such as *FoxL2*, *Esr2*, *Fst* (Fig. 3c), and genes expressed in granulosa cells

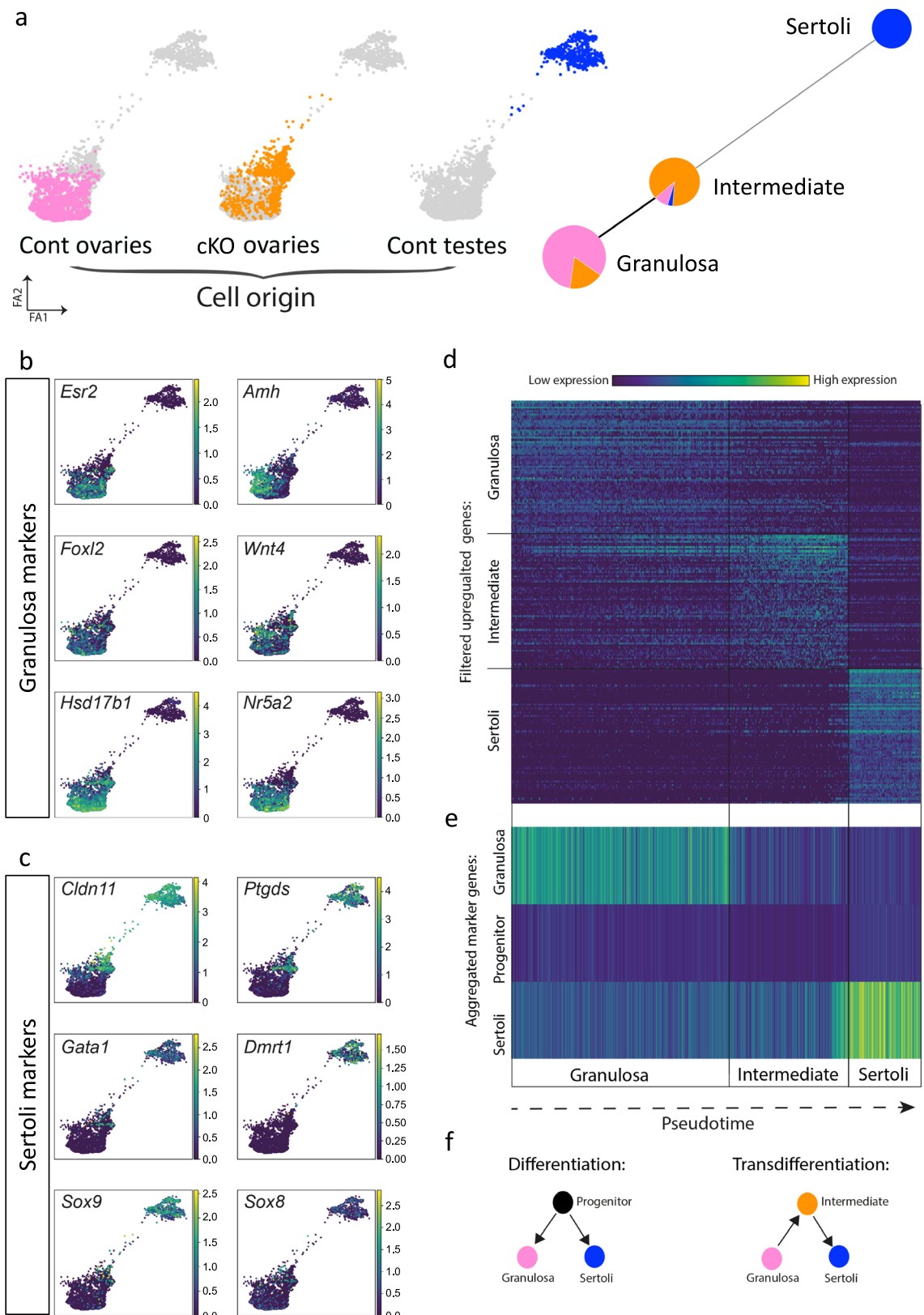

(Supplementary Fig. 11). As these genes were downregulated in *Trim28^cKO* ovaries, this suggests that TRIM28 and FOXL2 positively regulate major granulosa cell genes. For instance, *Wnt4*, which was downregulated in *Trim28^cKO* ovaries (Fig. 1c), displayed several TRIM28 and FOXL2 peaks in control ovaries (Supplementary Fig. 11). Conversely, *Rspo1*, which is upstream of *Wnt4* in the ovarian-determining cascade[1,31], was upregulated in *Trim28^cKO* ovaries (Fig. 1c). Analysis of the TRIM28/FOXL2 genomic profiles did not highlight any binding on *Rspo1* (Supplementary Fig. 11), suggesting that its regulation in the adult ovary is independent of TRIM28 and FOXL2. Moreover, in the absence of TRIM28, *Wnt4* expression seems to be independent from *Rspo1* expression level.

**Fig. 2 | scRNA-seq analysis of ovarian and testis supporting cells reveals an intermediate cell population during transdifferentiation. a** Force directed graphs showing the scRNA-seq results of adult *Trim28^cKO* ovarian supporting cells (orange), control granulosa (pink), and Sertoli cells (blue) (left). Each dot is one cell (coloured according to the sample of origin), and the distance between cells indicates their inferred transcriptional similarity. Leiden clustering divided the cells into three populations displayed using partition-based graph abstraction (right). Each node represents a cell cluster, and the proportion of *Trim28^cKO* and control granulosa and Sertoli cells is shown as a pie-chart on each node. The edges between nodes represent the neighbourhood relation among clusters with a thicker line

showing a stronger connection between clusters. **b, c** Gene expression of selected granulosa and Sertoli cell markers in the supporting cells analysed in **a**. Each dot corresponds to one cell from **a**, and gene expression level ranges from 0 (purple) to high (yellow). **d** Heatmap showing the expression level of the top filtered differentially expressed genes in the three cell clusters along the pseudo-time. See Table Data S3 for the full list of genes. **e** Heatmap showing the mean expression levels in the three cell clusters along the pseudo-time of several thousand genes from a previous study on the granulosa, supporting progenitor, and Sertoli cell lineages[29]. Source data are provided as a Source Data file. **f** Schematic illustrating the processes of differentiation and transdifferentiation.

Of note, 52% of the genes downregulated in *Trim28^cKO* ovaries interacted with TRIM28 and FOXL2 in control ovaries (Fig. 3d). Similarly, many testicular-specific genes upregulated in *Trim28^cKO* ovaries were bound by TRIM28 and FOXL2 (41%, 1189 of 2897), suggesting that TRIM28 and FOXL2 may have a repressive effect on the transcriptional activities of these genes in wild type ovary (Fig. 3d and Supplementary Fig. 12). For example, within the 2-Mb gene desert surrounding the *Sox9* gene, TRIM28 and/or FOXL2 peaks were in close proximity of some of the many enhancers implicated in gonadal *Sox9* expression regulation[32], and also in the proximal promoter and gene body (Fig. 3c, lower panel). Similarly, the distal upstream regions of *Dmrt1* and *Ptgds*, which are both upregulated in *Trim28^cKO* ovaries, displayed overlapping regions of TRIM28 and FOXL2 binding (Fig. 3c, lower panels), like other genes, such as *Cldn11* that is expressed in Sertoli cells and upregulated in *Trim28^cKO* ovaries (Supplementary Fig. 12).

We also analysed DNA motif enrichment for the binding sites of the major granulosa-specific transcription factors (FOXL2[4], RUNX[22] and ESR1/2[2]) in TRIM28 and FOXL2 ChIP-seq data, as previously described[8]. We observed significant enrichment for these motifs in regions bound by TRIM28 and FOXL2 in the ovary compared with regions bound by TRIM28 in bone marrow[33] and thymus[34] (Fig. 3e). This shows that in adult ovaries, both TRIM28 and FOXL2 bind to regions that display a genomic signature with binding sites for major ovarian-specific transcription factors.

To confirm that TRIM28 and FOXL2 colocalised on chromatin, we performed FOXL2 ChIP and selective isolation of chromatin-associated proteins (ChIP-SICAP) followed by mass spectrometry that provides only information relative to chromatin interactions[35]. We obtained a list of proteins colocalised with FOXL2 on ovarian chromatin that we ranked by their relative abundance. TRIM28 was amongst the top 20 FOXL2 interactors, confirming that it is recruited on chromatin regions very close to FOXL2 (Fig. 3f, left). It should be noted that TRIM28 has been recently shown[36] to interact with chromatin through two regions of the RBCC domains[37] (amino acids 298 to 305, and 349 to 366) and an intrinsically disordered region (amino acids 555 to 591). A gene ontology analysis of the protein list (that will be analysed and published elsewhere) showed that these proteins were mainly nuclear and chromatin factors, with only 3% of potential contaminants, demonstrating the technique specificity (Fig. 3f, right). These results are supported by a previous proteomic analysis of murine granulosa and pituitary-derived cell lines showing that TRIM28 and FOXL2 are engaged in common protein complexes[38]. Overall, the previous data on FOXL2[4] and our results show that in the ovary, TRIM28 and FOXL2 are implicated in the same pathway to maintain ovarian cell fate. On chromatin, this is achieved through their colocalization on regulatory regions of genes that control the granulosa and Sertoli cell fates. Our data suggest that the TRIM28 /FOXL2 pathway supports the granulosa cell fate by maintaining the ovarian identity and suppressing the testicular identity.

### Mutation of the SUMO-E3-ligase activity of TRIM28

TRIM28 acts as a SUMO-E3-ligase by interacting with the SUMO-E2 conjugating enzyme UBC9 (encoded by the *Ube2i* gene) via the Plant homeodomain (PHD) and can self-SUMOylate[39] (Fig. 4a). SUMOylation

is involved in transcriptional regulation and regulates positively or negatively the transcriptional activation capacity and/or stability of many transcription factors, such as FOXL2[40], ESR2[41], GATA4[42], PPARγ and RXR[43], and of many chromatin-associated proteins[44]. It is also an important histone modification (for review see[45]). Moreover, it has been reported that the SUMOylation status of transcription factors, such as NR5A1[46], and of androgen receptor[47] regulates their function in a tissue-specific fashion. Other proteins, such as PCNA[48], CDK9[49], NPM1/B23[50], IRF7[51], VPS34[52], α-synuclein, and tau[53], also are SUMOylated in a TRIM28-dependent manner. To study in vivo the role of TRIM28-dependent SUMOylation, we generated a point mutation in exon 13 of mouse *Trim28* within the PHD domain (C651F) that abrogates its SUMO-E3-ligase activity[50] (Supplementary Fig. 13). *Trim28^C651F/+* heterozygous mice reproduced normally and did not show any obvious phenotype. However, as we never obtained homozygous mutants when mating heterozygous animals, the homozygous *Trim28^C651F* mutation (termed *Trim28^Phd*) might be embryonic lethal, like *Trim28* ablation[54]. As heterozygous *Trim28 cKO* (*Nr5a1:Cre;-Trim28^flox/+*) mice have no phenotype (Fig. S6), we generated *Nr5a1:Cre;Trim28^C651F/flox* mice (*Trim28^Phd/cKO*). First, we showed that the TRIM28^C651F mutant protein was effectively produced and localised in the nucleus in *Trim28^Phd/cKO* mutant ovaries (Supplementary Fig. 14). RT-qPCR analysis of 8-week-old ovaries (Fig. 4b) showed that *Trim28* mRNA level in *Trim28^Phd/cKO* ovaries was intermediate between control (*Trim28^+/+*) and *Trim28^cKO* ovaries, confirming the presence of TRIM28^C651F transcripts. Moreover, ovarian- and testicular-specific genes (*FoxL2, Esr2, Wnt4, Hsd3b1, Ihh*, and *Sox9, Sox8, Dmrt1, Gata1, L-Pgds*, respectively) in *Trim28^+/+* and *Trim28^Phd/+* ovaries displayed similar expression levels, showing no dominant effect of the mutated allele. Conversely, in *Trim28^PHD/cKO* ovaries, ovarian genes were strongly downregulated, and testicular-specific genes were upregulated, like in *Trim28^cKO* ovaries.

This suggests that *Trim28^Phd/cKO* and *Trim28^cKO* ovaries display a similar phenotype. Next, we compared by immunofluorescence analysis, the expression of testis markers (SOX9, SOX8, and DMRT1) and of FOXL2 in *Trim28^Phd/cKO*, *Trim28^cKO*, and control ovaries. Like in *Trim28^cKO* ovaries, FOXL2 expression was undetectable, whereas we observed expression of the Sertoli cell markers SOX9, SOX8 and DMRT1 within structures organised in pseudo-tubules in *Trim28^Phd/cKO* ovaries (Fig. 4c). Histological analysis (Supplementary Fig. 15) also showed a similar tissue organisation in *Trim28^Phd/cKO* and *Trim2^cKO* ovaries. Altogether, these results indicate that the ovarian pathway maintenance in the adult ovary depends on the E3-SUMO ligase activity of TRIM28.

### TRIM28 mutants display a modified SUMOylation landscape

To determine whether the global SUMOylation level in the nucleus of granulosa cells was affected in *Trim28^Phd/cKO* and *Trim2^cKO* ovaries, we used a confocal microscopy quantitative analysis with anti-SUMO1 and -SUMO2/3 antibodies (called here SUMO2 because SUMO2 and 3 cannot be differentiated with antibodies). In both *Trim28^Phd/cKO* and *Trim28^cKO* ovaries, SUMO1 and particularly SUMO2 nuclear staining were decreased in ovarian somatic cells (Fig. 4d, left), as confirmed by fluorescence quantification (Fig. 4d, right). This shows that the absence of TRIM28 SUMO-E3-ligase activity in

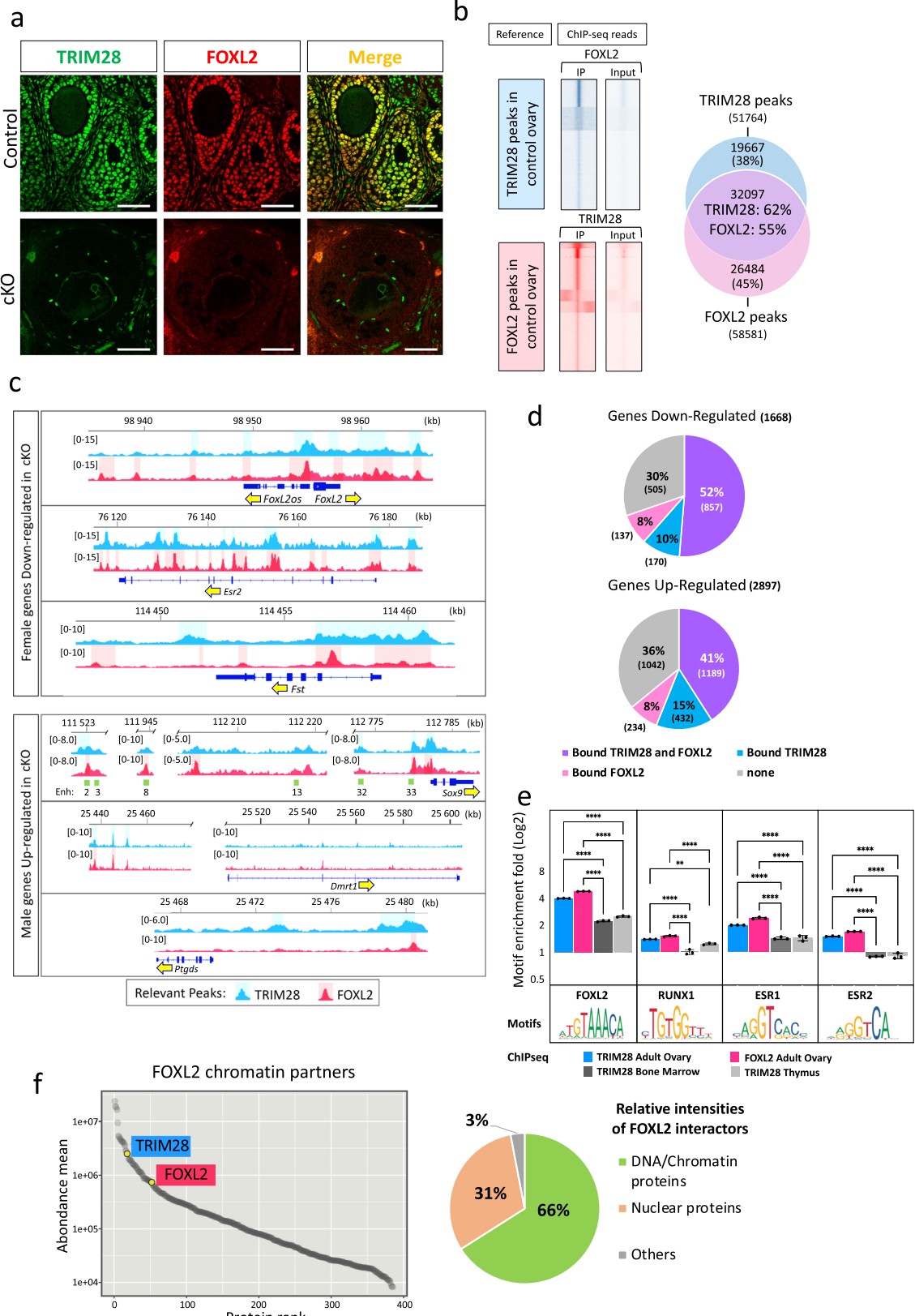

ovarian somatic cells decreased the nuclear level of SUMOylation, confirming the link between TRIM28 and this post-transcriptional modification in vivo.

As TRIM28 may SUMOylate some transcription factors or chromatin-associated proteins, we determined whether, in the two *Trim28* mutant mouse lines, the SUMOylation landscape was modified genome-wide. Quantitative SUMO1 and SUMO2 ChIP-seq analyses in adult *Trim28^{Phd/cKO}*, *Trim28^{cKO}* and control ovaries identified 249,760 chromatin regions that were SUMOylated by SUMO1 or SUMO2 in control ovaries.

As expected, in *Trim28^{cKO}* and *Trim28^{Phd/cKO}* ovaries, 7.3% and 5.2% of these peaks, respectively, displayed a significantly lower signal

**Fig. 3 | TRIM28 and FOXL2 act together on chromatin to maintain the ovarian pathway. a** TRIM28 and FOXL2 are co-expressed in the nucleus of most follicular granulosa cells in 4-month-old control ovaries and in cells with flat nucleus surrounding follicles (identified as steroidogenic theca cells). In *Trim28^{cKO}* ovaries, only few cells expressed FOXL2. Scale bar: 10 μm. At least three independent biological replicates were analysed, and the images presented are representative of all replicates. **b** Overlap between TRIM28 and FOXL2 genomic localisation in the adult ovary. Heatmaps in blue represent FOXL2 ChIP-seq and inputs reads mapped on TRIM28 peaks (±1 kb from the centre). Red traces represent TRIM28 ChIP-seq and inputs reads mapped on FOXL2 peaks. The Venn diagram on the right shows that 32,097 of the 51,764 TRIM28 peaks (62%) and of the 58,581 FOXL2 peaks (55%) overlap in control ovaries. **c** Examples of TRIM28 and/or FOXL2 peaks in/around genes the expression of which is altered in *Trim28^{cKO}* ovaries. Upper panel: ovarian-specific genes downregulated in *Trim28^{cKO}* ovaries (see also Supplementary Fig. 11). The *Foxl2* gene is represented with the co-regulated non-coding *Foxl2os* gene[106]. Lower panel: testicular-specific genes upregulated in *Trim28^{cKO}* ovaries (see also Supplementary Fig. 12). Green rectangles in the *Sox9* panel: open chromatin regions described in the embryonic gonads, 13

corresponds to Enhancer13 that is crucial for sex-determination[32]. Relevant ChIP-seq peaks are highlighted in light blue (TRIM28) and light red (FOXL2). Yellow arrows indicate the gene orientation. **d** Pie charts showing up- and downregulated genes in *Trim28^{cKO}* ovaries that are bound by TRIM28 and/or FOXL2. Genes are listed in Data S7. **e** Enrichment for binding motifs of transcription factors[8] involved in granulosa cell fate maintenance (FOXL2, RUNX1 and ESR1/2) in reads of TRIM28 and FOXL2 ChIP-seq of adult control ovaries (this study), and TRIM28 ChIP-seq of bone marrow[33] and of thymus[34]. *n* = 3 independent computational analyses. Bars are the mean ± SD. Ordinary one-way ANOVA with Tukey's multiple comparisons test. Adj *P* Val: for all motifs ****<0.0001. RUNX1: *=0.0186; **=0.0037; ***=0.0003. ESR1: ***=0.00020,0324 ESR2: ***=0.0005. More statistical data are in Source data file. Source data are provided as a Source Data file. **f** Left, plot showing enriched proteins, ranked by relative abundance, identified by FOXL2 ChIP-SICAP. Only significant proteins (>2-fold enrichment over No-antibody control, *n* = 2) are shown. TRIM28 was identified amongst the top 20 proteins found to interact with FOXL2. Pie-chart (right) shows the percentage of the relative intensities of FOXL2 chromatin partners, normalised to the total abundance of the enriched proteins.

(Log2 FC < −1, Adj *P* value >0.05) and we designated them as hypo-SUMOylated peaks (Fig. 5a, upper panel, and blue spots in Supplementary Fig. 16). The median size of these peaks was <1 kb (0.875 and 0.959 kb for *Trim28^{cKO}* and *Trim28^{Phd/cKO}*, respectively), but bigger than those obtained with TRIM28 or FOXL2 (0.775 and 0.532 kb, respectively). Of note, the number of hypo-SUMOylated peaks was higher in *Trim28^{cKO}* than *Trim28^{Phd/cKO}* ovaries (SUMO1 + SUMO2 peaks: 18,338 versus 12,972), suggesting that the C651F mutation may not completely abolish TRIM28 E3-ligase activity, although it induces granulosa-to-Sertoli cell transdifferentiation, as indicated by the similar phenotype of the two mutants.

Quantification of SUMO1 and SUMO2 ChIP-seq reads that mapped to hypo-SUMOylated peaks (Fig. 5a) showed that they were markedly decreased in *Trim28^{cKO}* and *Trim28^{Phd/cKO}* samples (Fig. 5a, upper panel in blue). Moreover, quantification of TRIM28 ChIP-seq reads from control ovaries showed that they mapped strongly to these regions (Fig. 5b, box plots in blue). This shows that in control ovaries, TRIM28 occupies chromatin regions that are hypo-SUMOylated in *Trim28^{cKO}* ovaries, strongly implying that TRIM28 is the E3-ligase responsible of their SUMOylation in the adult ovary (either auto-SUMOylation or SUMOy-lation of transcription factors located near TRIM28 on chromatin). For example, many hypo-SUMOylated regions in *Trim28^{cKO}* ovaries were occupied by TRIM28 and FOXL2 in control ovaries (Supplementary Fig. 17), suggesting that FOXL2 might be a TRIM28 substrate. Moreover, it has been reported that FOXL2 is SUMOylated in ovarian cell lines[40,55–57] where this modification might promote its stabilisation[40,56,57]. Similarly, ESR2 stability is regulated by SUMOylation[41]. Analysis of recently published ESR2 ChIP-seq data[58] also showed that ESR2 peaks overlapped with the hypo-SUMOylated peaks of our mutants, but to a lesser extent than what was observed for FOXL2 (Supplementary Fig. 17). RUNX1 is another transcription factor involved in the maintenance of the fetal ovarian fate that shares with FOXL2 a substantial number of genomic targets[22]. Due to the absence of publicly available RUNX1 ChIP-seq data in adult ovaries, we performed SUMOylation assays in cells transfected with wild-type TRIM28 or the PHD mutant. We observed that TRIM28 wild type, but not the PHD mutant induced SUMOylation of both FOXL2 and RUNX1 (Supplementary Fig. 18), suggesting that both factors are potential substrates of TRIM28 E3-ligase activity.

However, TRIM28-dependent SUMOylation of transcription factors might also occur before their interaction with chromatin because only a fraction (33–45%) of hypo-SUMOylated regions in *Trim28^{cKO}* and *Trim28^{Phd/cKO}* ovaries were occupied by TRIM28 in control ovaries (Supplementary Fig. 17). We also found a substantial number of SUMO1 or SUMO2 peaks with a significantly stronger signal in *Trim28^{cKO}* or *Trim28^{PHD/cKO}* than control ovaries (Log2 FC > 1, Adj *P* val >0.05) that we designated as hyper-SUMOylated (Fig. 5b, upper panel, and red spots in

Supplementary Fig. 16). ChIP-seq read quantification showed that in *Trim28^{cKO}* and *Trim28^{Phd/cKO}* ovaries, hyper-SUMOylation (SUMO1 and SUMO2) occurred de novo on regions that were fewer SUMOylated in control ovaries (Fig. 5a, lower red panels). Moreover, quantification of TRIM28 ChIP-seq reads in control ovaries showed that these hyper-SUMOylated regions were poorly occupied by TRIM28 (Fig. 5b, box plots in red), unlike hypo-SUMOylated regions (Fig. 5b, box plots in blue). In agreement, peak analysis showed nearly no overlap between hypo- and hyper-SUMOylated regions in both mutants (Supplementary Fig. 19). These hyper-SUMOylated peaks might be the signature of Sertoli cell-specific transcription factors expressed in transdifferentiated granulosa cells. To test this hypothesis, we analysed SOX9 and DMRT1 ChIP-seq data during granulosa-to-Sertoli cell transdifferentiation induced by ectopic DMRT1 expression in the ovary[58]. We found that in both *Trim28^{cKO}* and *Trim28^{Phd/cKO}* ovaries, 14 to 18% of hyper-SUMOylated peaks overlapped with DMRT1 peaks, while 3 to 5% overlapped with those of SOX9 (Fig. S20). Although more experiments are required to confirm that DMRT1 is SUMOylated, our analysis shows that some hyper-SUMOylated peaks are effectively occupied by DMRT1 and SOX9 during adult reprograming of granulosa-to Sertoli cells.

Our results showed that downregulation of the ovarian pathway in *Trim28^{cKO}* and *Trim28^{Phd/cKO}* ovaries allows the activation of another pathway, inducing the de novo SUMOylation of distinct chromatin regions, possibly related to the activated testicular genes. Yet, the RNA-seq analysis of *Trim28^{cKO}* ovaries (Data S1) did not highlight the upregulation of any testicular-specific E3-SUMO ligase (e.g., proteins of the PIAS family). This suggests that such ligases are expressed also in granulosa cells.

Analysis of the list of hypo- and hyper-SUMOylated genes highlighted a strong correlation between the very similar phenotypes of the two mutants and gene SUMOylation. Specifically, 5,082 and 4,056 genes were hypo- and hyper-SUMOylated, respectively, in both *Trim28^{cKO}* and *Trim28^{Phd/cKO}* ovaries (Supplementary Fig. 21a). Some genes showed a mixed SUMOylation pattern (both hypo- and hyper-SUMOylation peaks) (Supplementary Fig. 21b), suggesting a more complex regulation. However, most genes were strictly hypo- (74%) or hyper- (75%) SUMOylated, indicating that they belong to distinct pathways.

Next, we analysed the SUMOylation status of the genes identified as upregulated or downregulated in *Trim28^{cKO}* ovaries by RNA-seq. Among the 1669 downregulated genes (Fig. 5c, upper pie-chart), the genes displaying SUMOylation variations were preferentially hypo-SUMOylated (26%), while a minority were hyper-SUMOylated (9%) or both hypo- and hyper-SUMOylated (8%). Ovarian-specific genes that were downregulated in *Trim28^{cKO}* ovaries (*Cyp11a1, Esr2, Foxl2, Fst,* and *Hsd3b1*) displayed hypo-SUMOylated peaks in *Trim28^{cKO}* and

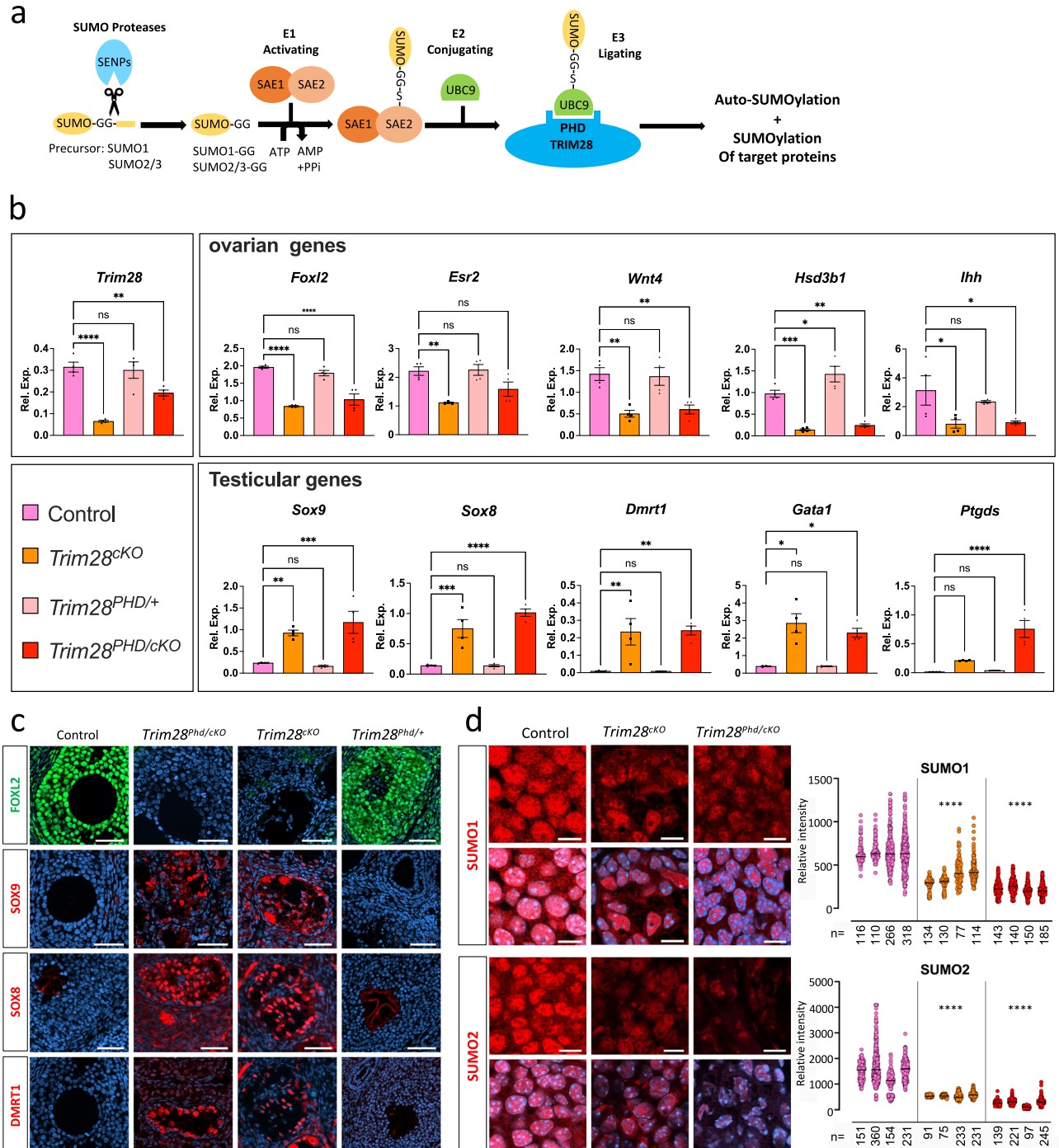

**Fig. 4 | Loss of TRIM28 SUMO-E3-ligase activity in granulosa cells phenocopies** *Trim28* **conditional knock-out. a** Schematic of the SUMO pathway with TRIM28 E3-SUMO ligase activity. After proteolytic maturation by sentrin-specific proteases (SENPs), SUMO C-terminus is activated by the heterodimeric SUMO-activating enzyme E1 (SAE1/SAE2), and then transferred to a cysteine of E2 (UBC9). Subsequently, the E3 ligases (TRIM28) transfer SUMO from E2 to a lysin residue(s) of target proteins. SUMO2 and 3 diverge by only one residue, making them indistinguishable by antibodies, thus they are currently referred to as SUMO2. **b** RT-qPCR analysis of ovarian- and testicular-specific genes in 8-week-old *Trim28^cKO*, *Trim28^Phd/cKO*, *Trim28^Phd/+*, and control ovaries. Bars are the mean ± SEM, *n* = 5 animals (gonad pairs). *P:* < 0.0001 (****), *0.0002(***), 0.0021(**), 0.032(*)* (Ordinary one-way ANOVA with Dunnett's multiple comparisons test). Details of the statistical analysis are provided in the Source data file. Source data are provided as a Source Data file. **c** FOXL2 is expressed in control and *Trim28^Phd/+* ovaries, but not in *Trim28^Phd/cKO* and

*Trim28^cKO* ovaries. Like in *Trim28^cKO* ovaries, SOX9, SOX8 and DMRT1 are expressed in pseudo-tubules of *Trim28^Phd/cKO* ovaries, but not in control and *Trim28^Phd/+* ovaries. Protein (green or red) is merged with DNA stains (blue). Scale bar: 50 μm. **d** Confocal microscopy shows strong SUMO1 and 2 nuclear staining in granulosa cells of control ovaries. The staining intensity is markedly decreased in *Trim28^cKO* and *Trim28^Phd/cKO* ovaries. SUMO1/2 staining is merged with DNA staining. Scale bar: 20 μm. Right panels: quantification of SUMO1 and SUMO2 signal intensity relative to DNA staining. For the three conditions (control and mutants) each column represents one experiment, *n* represents the number of cells analysed. *P:* < 0.0001 (****), *0.0002(***), 0.0021(**), 0.032(*)*(two-way ANOVA with Dunnett's multiple comparisons test). Details of the statistical analysis are provided in Source data file. Source data are provided as a Source Data file. For the immunofluorescences, at least three independent biological replicates were analysed, and the images presented are representative of all replicates.

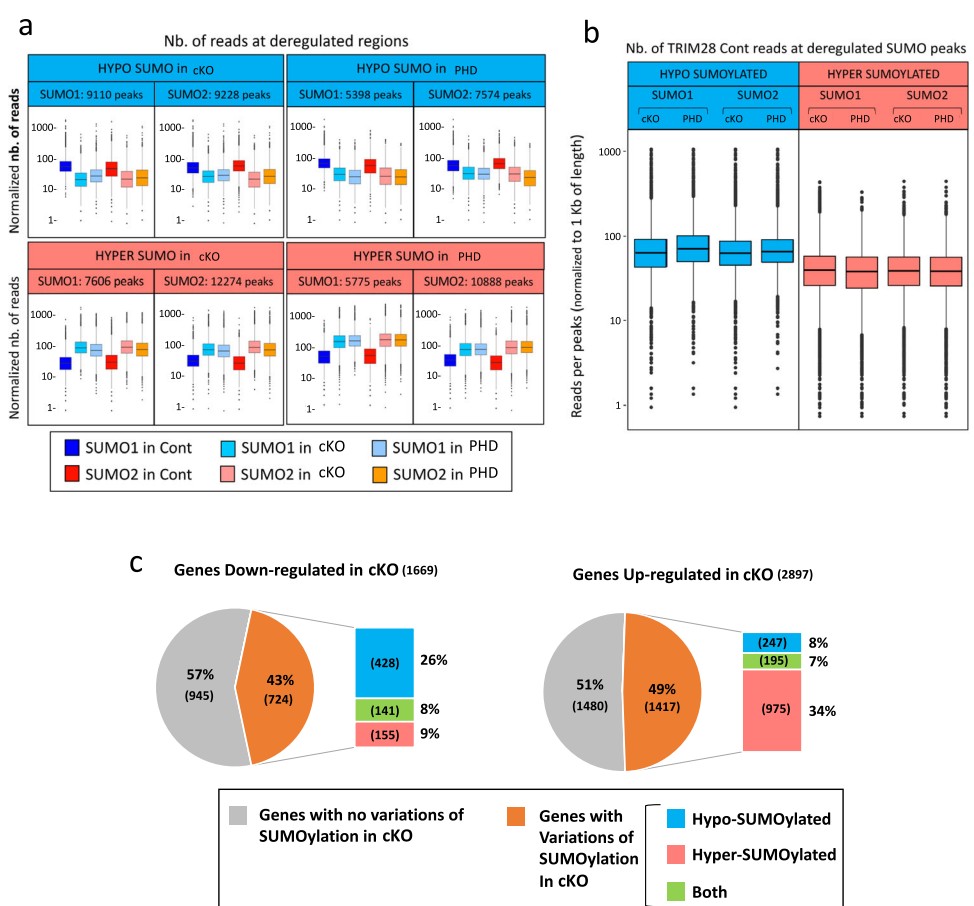

**Fig. 5 | Genome-wide SUMOylation changes in *Trim28^cKO* and *Trim28^Phd/cKO* ovaries. a** Normalised quantification of SUMO1 and SUMO2 ChIP-seq reads from control (Cont), *Trim28^cKO* (cKO) and *Trim28^Phd/cKO* (PHD) ovaries mapped on deregulated regions: peaks significantly *decreased* (Log2 Fold-Change <1; hypo-SUMOylated; blue), and *increased* (Log2 Fold-Change >1; hyper-SUMOylated; red) in *Trim28^cKO* and *Trim28^Phd/cKO* ovaries compared with controls. The number of peaks analysed for each condition is reported on upper (blue or light red) of each chart. **b** Normalised quantification of TRIM28 ChIP-seq reads from control (±1 kb from the centre) at SUMO1 and SUMO2 hypo-SUMOylated peaks (blue box plots) and SUMO1 and 2 hyper-SUMOylated peaks (red box plots). cKO: *Trim28^cKO*. PHD: *Trim28^Phd/cKO*. For box plots, the centre line corresponds to the median. The number

of peaks analysed is the same as reported in Fig. 5a. For **a** and **b**, the central rectangle spans the first quartile (Q1) to the third quartile (Q3) (also called IQR for interquartile range). The upper whisker extends from the hinge to the largest value no further than $1.5 \times IQR$ from the hinge ($Q3 + 1.5 \times IQR$). The lower whisker extends from the hinge to the smallest value at most $1.5 \times IQR$ of the hinge ($Q1 - 1.5 \times IQR$). Number of libraries: TRIM28 ChIP-seq, $n = 1$; for SUMO1 and SUMO2 (from control, *Trim28^cKO* and *Trim28^Phd/cKO*) ChIP-seq $n = 2$. Each library was prepared from ovaries of six different animals. **c** Pie charts showing that in *Trim28^cKO* ovaries, down-regulated genes with SUMOylation changes are preferentially hypo-SUMOylated, while upregulated genes with SUMOylation changes are preferentially hyper-SUMOylated. Number of genes are between brackets. Genes are listed in Data S7.

*Trim28^Phd/cKO* samples for both SUMO1 and SUMO2 (Fig. 6, upper panels, and Supplementary Fig. 22), where TRIM28 and FOXL2 are bound in control (Fig. 3c).

Conversely, among the testicular genes upregulated in *Trim28^cKO* ovaries (Fig. 5c, lower pie-chart), genes showing SUMOylation variations were preferentially hyper-SUMOylated (34%), and only 8% and 7% were hypo-SUMOylated and both hyper- and hypo-SUMOylated, respectively (examples in Fig. 6, lower panel, and Supplementary Fig. 23). The key testicular-specific genes *Sox9* and *Dmrt1* that are strongly repressed in granulosa cells showed a mixed SUMOylation pattern in the mutants. At the *Sox9* locus, we observed a mixed hypo- and hyper-SUMOylation pattern in the large regulating region upstream of the gene body: four hyper-SUMOylated peaks and three hypo-SUMOylated peaks in the proximity and along the enhancers 13, 22 and 26[32]. Similarly, in the *Dmrt1* gene, we detected two hyper-SUMOylated regions, one in the gene body and the other upstream, and one hypo-SUMOylated region. These complex SUMOylation patterns could reflect the need of strict regulation because the expression of these two genes must be silenced in granulosa cells. By contrast, *Sox8* and *Ptgds* (like the testicular genes presented in Supplementary Fig. 23) displayed only hyper-SUMOylation peaks, suggesting that

SUMOylation might reflect only their transcriptional activation. Another example is *Cldn11*, one of the earliest Sertoli-specific genes (Fig. 2c, Supplementary Fig. 10). We detected TRIM28 and FOXL2 peaks at four different regions of the *Cldn11* genomic locus (Supplementary Fig. 12), likely to repress its expression. However, the most upstream of these regions, which is an open chromatin region in embryonic gonads[59], was hyper-SUMOylated in the cKO and PHD mutants (Supplementary Fig. 23). Therefore, upon the disappearance of TRIM28 and/or FOXL2 in mutants, some transcription factors might have access to this potential enhancer, to activate the *Cldn11* gene.

Overall, the TRIM28 E3-ligase controls the maintenance of granulosa cell fate via the specific SUMOylation of ovarian genes. In its absence, a distinct pathway takes place, leading to the hyper-SUMOylation of some Sertoli cell-specific genes that are correlated with their activation.

## Discussion
This study shows that *Trim28* plays a central role in the postnatal maintenance of the ovarian somatic cell fate. Upon *Trim28* loss in fetal pre-granulosa cells, differentiated granulosa cells are reprogrammed, after birth, into Sertoli cells through a previously undescribed

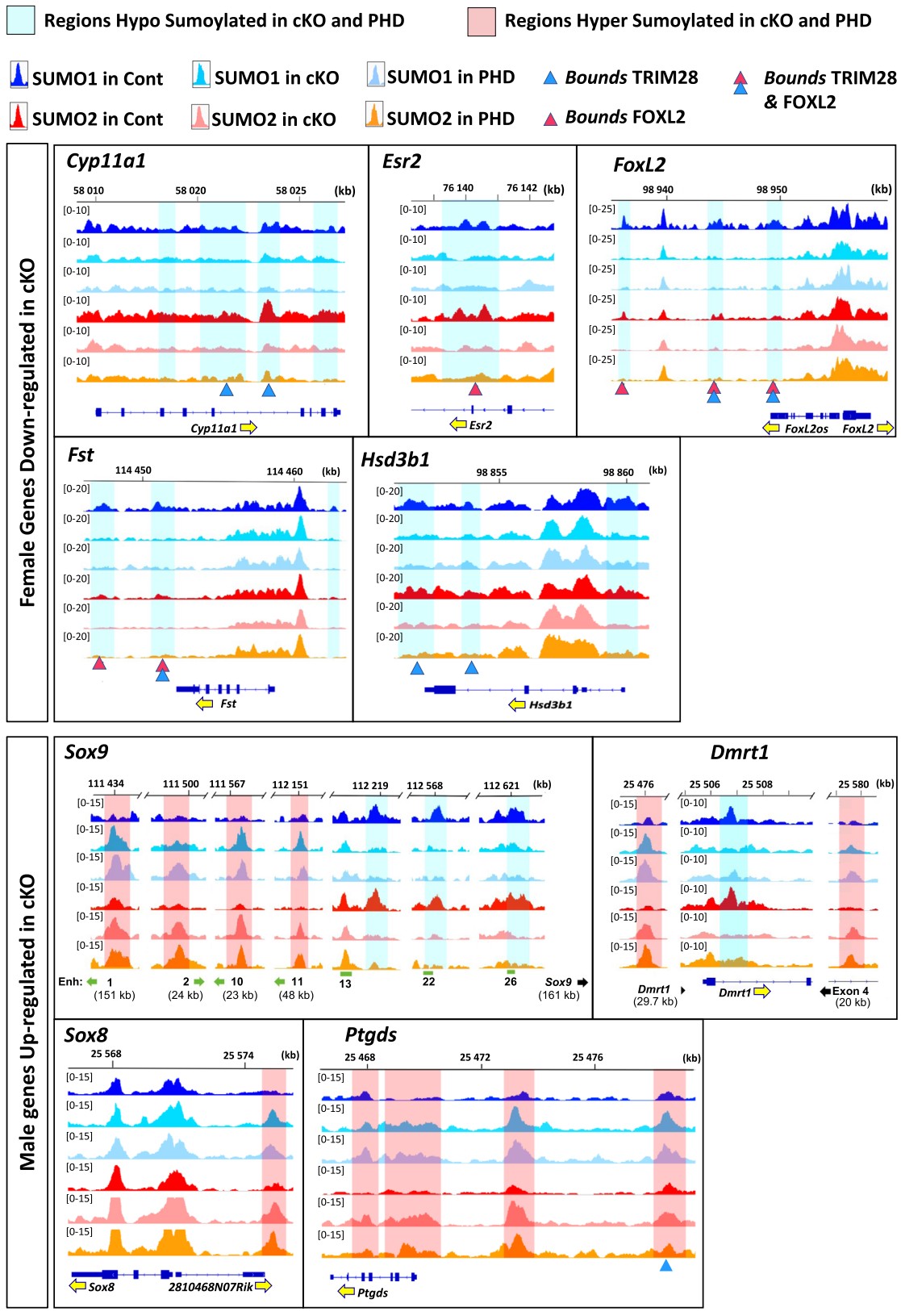

**Fig. 6 | Examples of SUMOylation status (SUMO1 and 2) in control and mutant ovaries of genes the expression of which is altered in _Trim28^cKO_ ovaries.** Upper panel: ovarian-specific genes downregulated in _Trim28^cKO_ ovaries. Lower panel: testicular-specific genes upregulated in _Trim28^cKO_ ovaries. Cont: control. cKO: _Trim28^cKO_. PHD: _Trim28^Phd/cKO_. Yellow arrows indicate the gene orientation. Light blue and red, regions significantly hypo-SUMOylated and hyper-SUMOylated, respectively, in mutants. Blue and red triangles represent the centre of TRIM28 and FOXL2 peaks respectively (see supplementary Fig. 11 and 12). Green rectangles in the _Sox9_ panel: putative enhancers and Enhancer13 previously described[32]. Green arrows indicate the distance relative to the putative enhancers.

intermediate cell type. Therefore, granulosa cells do not dedifferentiate into embryonic progenitors, but acquire a different cell state in which neither ovarian nor testicular master genes are expressed. Moreover, our scRNA-seq and immunofluorescence data confirmed that transdifferentiation is the only possible mechanism of sex reversal and excluded the de novo generation of Sertoli cells concomitantly to granulosa cell disappearance (e.g., due to massive apoptosis). Of note, during somatic sex reprogramming in *Foxl2*[-/-] adult ovaries[4], for ~1 day following the disappearance of FOXL2 expression, SOX9 cannot be detected, suggesting a similar intermediate step as observed in the present work.

Unexpectedly, structural genes of Sertoli cells, such as *Cldn11*, were upregulated before key genes encoding testicular transcription factors, such as *Sox9* and *Dmrt1*. This suggests that the onset of transdifferentiation might not occur through the activation of a single master gene, such as *Sox9* or *Dmrt1*, but through the global de-repression of the testicular-specific transcriptome. Our observation that TRIM28 is a co-factor of FOXL2 on chromatin supports this hypothesis. In the absence of functional TRIM28, FOXL2 would progressively lose its capacity to repress the testicular pathway, leading to a global de-repression of Sertoli cell genes. The potential role of SOX8 needs to be better investigated. Immunofluorescence, RT-qPCR, and scRNA-seq experiments showed that *Sox8* is upregulated before *Sox9* and *Dmrt1* in *Trim28*[cKO] ovaries. However, as SOX8 has a weak trans-activation capacity[23], the transdifferentiation process might be accelerated by de-repression of testicular pathway master genes (*Sox9* and *Dmrt1*). A recent study has shown that DMRT1 acts as a pioneering factor required by SOX9 for the optimal activation of its target genes[58]. In our case, the engagement in the testicular pathway might be partial until *Dmrt1* is fully activated. Additional genetic experiments, using double *Trim28* and *Sox8*, *Sox9*, or *Dmrt1* knock-out lines are required to answer this question.

At the organ level, transdifferentiation is first completed in the medulla and then extends to the cortical region. At week 8 post-partum, mutant ovaries displayed medullar pseudo-tubules and cortical follicles: a two-step process also observed in mice where both oestrogen receptors were knocked out[60]. Interestingly, medullar granulosa cells are mostly derived from bipotential precursors in which primary sex-determination occurs at 11.5 dpc and that are integrated in follicles at puberty[61,62]. Conversely, cortical follicle pre-granulosa cells are generated mainly by the celomic epithelium from 13.5 dpc until birth and sustain fertility[63,64]. This suggests that bipotential precursor-derived medullar granulosa cells might be more sensitive to the effect of *Trim28* absence/mutation.

An important finding of our study is the role of TRIM28-dependent SUMOylation in the maintenance of granulosa cell fate. Previous work showed that global SUMOylation of chromatin-associated proteins has a key role in the stabilisation of somatic and pluripotent states[65]. Here, we found that TRIM28-dependent SUMOylation, which represents less than 10% of the whole SUMOylation landscape, is sufficient to prevent adult sex reversal. TRIM28 induces relatively sharp peaks of SUMOylation on chromatin (<1 kb), unlike the large peaks of histone modifications. This might reflect SUMOylated transcription factors. Therefore, a central question is the nature of TRIM28 targets. As TRIM28 can self-SUMOylate[39], a large number of hypo-SUMOylated peaks in *Trim28*[cKO] and *Trim28*[Phd/cKO] ovary samples may represent TRIM28 SUMOylation; this was confirmed by the overlap between these peaks and TRIM28 peaks in control ovary samples. Similarly, many FOXL2 peaks overlapped with hypo-SUMOylated peaks in *Trim28*[cKO] and *Trim28*[Phd/cKO] ovary samples, and our in vitro data showed that TRIM28 SUMOylates FOXL2. It was previously shown that FOXL2 SUMOylation leads to its stabilisation[40,57]. This could explain why in *Trim28*[cKO] ovaries, FOXL2 is undetectable, although the transcript is still present. Indeed, the lack/reduced SUMOylation of FOXL2 might contribute to its progressively decrease/

loss in postnatal ovary, leading to transdifferentiation. However, many hypo-SUMOylated peaks did not overlap with TRIM28 or FOXL2, indicating that other transcription factors or chromatin-associated proteins display TRIM28-dependent SUMOylation, particularly FOXL2, ESR1/2, and RUNX1 that are involved in ovarian maintenance[2,22]. We also observed that RUNX1 is SUMOylated by TRIM28 in vitro and that ESR1 transcriptional activity[66] and ESR2 stability[41] are positively regulated by SUMO-conjugation. Moreover, genome-wide, ESR2 also binds to hypo-SUMOylated peaks, but in a smaller proportion than FOXL2. In the absence of TRIM28, these transcription factors and FOXL2 might lose their capacity to maintain the ovarian programme.

Our data support the hypothesis that a TRIM28-dependent programme of SUMOylation maintains the transcription of ovarian genes and represses genes involved in Sertoli cell fate. These results challenge the dominant view for many years according to which SUMOylation only represses transcription. Our findings and a recent study on the control of adipogenesis by SUMOylation[43] suggest a more complex scenario: SUMOylation of important cell regulators (e.g., transcription factors) via a specific SUMO-E3-ligase (e.g., TRIM28) might regulate a complete transcriptional programme through activation or repression of target genes. So far, no function other than E3-SUMO ligase activity has been attributed to the PHD domain of TRIM28. We cannot exclude that this domain may have another function even if our genome-wide analyses are strongly in favour of a major role of it E3-ligase activity.

As the *Trim28*[cKO] and *Trim28*[Phd/cKO] ovarian transcriptomes displayed a strong masculinisation, we also observed activation of a de novo pattern of chromatin SUMOylation (i.e., hyper-SUMOylated peaks) that we attributed to the testicular pathway and that is catalysed by a still unknown E3-ligase. These hyper-SUMOylated peaks might represent SUMOylated Sertoli-specific transcription factors, such as DMRT1 or SOX9 that can be SUMOylated[67]. Importantly, by analysing ChIP-seq data obtained by Lindeman and colleagues in ovarian reprograming experiments[58], we found that a substantial amount of the hyper-SUMOylated peaks from our results colocalised with DMRT1 peaks and to a lesser extent with SOX9 peaks. This shows that both transcription factors are present in hyper-SUMOylated regions and might be SUMOylated (or their partners) independently of TRIM28. SUMO proteomic approaches should answer these questions about hypo- and hyper-SUMOylated peaks.

Altogether, our findings suggest a multi-step model. First, in the absence of TRIM28, FOXL2 that colocalises on chromatin with TRIM28 would lose its capacity to maintain the expression of granulosa cell-specific genes. Granulosa cells would differentiate into an intermediate state where they express non-sex-specific markers. Second, this would lead to the de-repression of some Sertoli cell-specific genes, such as *Sox8* or *Cldn11*, allowing progressively the induction of strong activators of the Sertoli cell pathway, such as *Dmrt1*. To confirm this model, we need now to generate mice lacking both *Sox8*, *Sox9* or *Dmrt1* and *Trim28*.

Unlike its role in granulosa cells, TRIM28 is not required for the maintenance of adult Sertoli cells where it is involved in their crosstalk with germ cells[20] and also in SUMOylation[68]. However, as we could not completely abolish TRIM28 protein expression in pre-granulosa cells before 13.5 dpc, we cannot exclude a role in primary sex-determination that occurs at 11.5 dpc. Indeed, in vitro studies have shown that the testis-determining factor SRY, through its interaction with a KRAB-0 protein[69], might recruit TRIM28 on chromatin to repress ovarian genes[70]. Therefore, more genetic experiments are required to delete *Trim28* using Cre drivers that work earlier, as previously described for *Gata4*[71].

TRIM28 is an important player in ovarian physiology and therefore, might also have a potential role in genetic diseases causing reproductive disorders. TRIM28 has been recently identified as a tumour suppressor in Wilms' tumour, a common paediatric kidney

malignancy (reviewed in[72]). However, no *TRIM28* mutation has been described so far in patients with reproductive disorders, such as primordial ovarian insufficiency[73], and in patients with disorders of sexual development (Dr Ken McElrevey personal communication). Besides genetic alterations, environmental factors, such as drugs or chemicals may also interfere with the SUMO-E3-ligase activity of TRIM28, and this could, in turn, perturb ovarian function and fertility.

# Methods

This study complies with all relevant ethical regulations and has been approved and funded by "Agence Nationale de la Recherche" (ANR-16-CE14-0020-SexMaintain to F.P)

## Generation of mutant mice

Animal care and handling were according to the "Reseau des Animaleries de Montpellier" (RAM) guidelines. All mouse lines were kept on a mixed 129SV/C57BL6/J background. Animals were maintained on a 12-hour light/dark cycle. The colony was maintained at a temperature of 22-24 °C, with humidity at 30–70%. Mice of both sexes were used. Ages: 4 dpp, 20 dpp, 2, 4 and 7 months. Embryos of E13.5 and E18.8 were also used. A number of animals are described as "n" in figure legends and for histological experiments, at least three biological replicates were analysed. The *Nr5a1:Cre* line[21] and its use to generate conditional knock-outs in ovarian granulosa cells were described previously[22] and provided by late Dr. Keith Parker. The *Trim28* conditional allele was described previously[54] and provided by Dr. Florence Cammas. To generate granulosa *Trim28* conditional knock-out mice, mice carrying *Trim28* loxP-flanked alleles (flox) (*Trim28^flox/flox^*) were crossed with mice bearing the *Nr5a1:Cre* transgene to generate *Trim28^flox/+^*; *Nr5a1:Cre* mice. These mice were then crossed with *Trim28^flox/flox^* mice to generate *Trim28^flox/flox^*; *Nr5a1:Cre* female mice; these mice were referred to as *Trim28^cKO^* null mutants. These crosses also generated *Trim28^flox/flox^* mice without the *Nr5a1:Cre* transgene and *Trim28^flox/+^*; *Nr5a1:Cre* mice; both genotypes did not show any histological defect and were thus referred to as control mice.

The *Trim28^PHD/+^* mutant mouse line was established at the Mouse Clinical Institute - Institut Clinique de la Souris, Illkirch, France (http://www.mci.u-strasbg.fr). *Trim28^Phd/+^* embryonic stem cells (see Supplementary Fig. 11 for the generation of the targeted C651F mutation in exon 13 of *Trim28*) were used to derive *Trim28^PHD/+^* mice. To generate granulosa knock-in mice that express only the mutant TRIM28 protein, *Trim28^Phd/+^* mice and mice bearing the *Nr5a1:Cre* transgene were first crossed to produce *Trim28^Phd/+^*: *Nr5a1:Cre* mice. These mice were then crossed with *Trim28^flox/flox^* mice to generate *Trim28^Phd/flox^*: *Nr5a1:Cre* mice; these mice were referred to as *Trim28^Phd/cKO^* knock-in mutants or *PHD* mutants. These crosses also generated *Trim^Phd/flox^* mice without the Nr*5a1:Cre* transgene, *Trim28^Phd/+^*; N*r5a1:Cre* mice, and *Trim28^flox/+^*; Nr*5a1:Cre* mice that did not have any histological defect, and were referred to as control mice. No *Trim28^Phd/Phd^* homozygous mouse was obtained when *Trim28^Phd/+^* mice were crossed. This suggests that the homozygous PHD mutation may be embryonic lethal, as already observed for the TRIM28^HP1box^ mutation[20].

*Trim28* floxed allele was genotyped with primers surrounding the loxP insertion site[54]: 5′-GGAATGGTTGTTCATTGGTG-3′ and 5′-ACCTT GGCCCATTTATTGATAAAG-3′. The wild-type allele gives a PCR product of 152 bp, and the floxed allele of 180 bp.

The *Trim28^Phd^* allele was genotyped with primers that surround the mutation in exon 13 (5′-AAGCCTGTGTTGATGCCTCT-3′ and 5′-CTTCAGCTACTGGGCCACAC-3′) and that give a PCR product of 513 bps in the wild type allele. The mutation introduces a site for the restriction enzyme *Afe*I that cuts the sequences amplified by the primers in two fragments of 240 and 273 bp, respectively. The *Phd* allele gives also a PCR product of 180 bps with the primers used for the *Trim28* floxed allele.

The Nr*5a1:Cre* transgene was genotyped using the 5′-TCGG GGTTTTGTTCTCAGAC-3′ and 5′- ATGTTTAGCTGGCCCAAATG-3′ primers that give a PCR product of around 500 bp.

PCR conditions for all genotyping were: 94 °C for 30 sec; 55 °C for 30 sec, 72 °C for 30 sec, 30 cycles.

## ChIP-seq analysis

Dissected ovaries were snap-frozen and stored at −80°C. Frozen ovaries were crushed in liquid nitrogen using a mini-liquid nitrogen-cooled mortar (Bel-Art ref H37260-0100). Powdered tissue samples were immediately fixed in PBS/1% formaldehyde at room temperature for 20 min. Chromatin shearing was performed using a qSonica Sonicator. Chromatin immunoprecipitation and sequencing libraries were prepared using the Low Cell ChIP-Seq Kit, the Next Gen DNA Library Kit that contains molecular identifiers (MIDs) used to remove PCR duplicates from sequencing data, and the Next Gen Indexing kit (Active motif, ref 104895). For peak calling, the peak caller that best fitted our experimental design and data was chosen and followed the ENCODE consortium recommendations[74] (see also: https://www.encodeproject.org/pipelines/).

For TRIM28 and FOXL2 ChIP-seq: Each ChIP-seq library was prepared from a pool of three independent immunoprecipitations (IP), and each IP was prepared with ovaries of two different animals. Reads were mapped to the *Mus musculus* genome (assembly mm10) using Bowtie[75] v1.0.0 with default parameters except for "-p 3 -m 1 −strata −best −chunkmbs 128. BAM files were sorted using SAMtools[76] v0.1.19. The tool rmDupByMids.pl provided by Active Motif was then used to remove duplicated reads. Peaks were called using MACS2[77] with default parameters except for "-g mm -f BAM −broad −broad-cuto 0.1 −keep-dup all".

For SUMO1 and SUMO2 ChIP-seq: Two independent ChIP-seq libraries were prepared. For each library three independent IPs were pooled, each IP prepared from ovaries of two different animals. Reads were mapped to the *Mus musculus* genome (assembly mm10) using Bowtie[75] v1.0.0 with default parameters except for "-p 3 -m 1 −strata −best". BAM files were sorted using SAMtools[76] v0.1.19. The tool rmDupByMids.pl provided by Active Motif was then used to remove duplicated reads. For peak calling data were analysed using the Encode ChIP-seq pipeline v1.3.6[74]. Briefly, the pipeline ran quality controls and called peaks with spp v1.15.5[78]. Spp v1.15.5 was run through the ENCODE pipeline v1.3.6 using the following parameters "-npeak=300000 -fdr=0.01". The fragment length parameter "-speak" was set according to the value estimated by the ENCODE pipeline. Reproductible peaks were kept after the IDR analysis was run. Peaks were annotated relative to genomic features using Homer v4.11.0[79] with Ensembl v92 annotations. Regions differentially bound between conditions were detected as follows. For all proteins of interest, all detected peaks were combined to merge all peaks (249760 peaks) using Bedtools merge v2.26.0. Finally, read counts were normalised across libraries using the method proposed by[80]. Comparisons were performed using the method proposed by[81] implemented in the DESeq2 Bioconductor library (DESeq2 v1.6.3). The resulting *p* values were adjusted for multiple testing using the Benjamini and Hochberg method[82]. The following thresholds were used to select differentially bound regions: adjusted *p* value ≤0.05, |log2 Fold-Change|≥1. Heatmap analyses were performed with SeqMINER v1.3.3g[83]. On all graphs, the presented data are a sub-sample of the initial alignment data (10 million) to make the enrichments in reads comparable. For box plots, all SUMO peaks were normalised to 1Kb. The number of reads per peak was calculated using bedtools intersect v2.26.0. The graphs are made with R v4.1.1 scripts. Peak overlap was analysed using the intersects tool from the Bedtools package[84] in Galaxy. Motif enrichment was assessed as previously described[8] using the matrix-scan tool (*P* value 1e-4) from RSAT{Nguyen, 2018 #17620 (http://rsat.sb-roscoff.fr) and

### ChIP-SICAP, mass spectrometry and data analysis

Ovaries from 8-week-old C57BL/6 mice were dissected in PBS, snap-frozen, and stored at −80 °C. Chromatin was prepared using a modified version of the Active Motif High Sensitivity Chromatin Prep Kit. Frozen tissues were pulverised in liquid nitrogen using the Bel-Art™ SP Scienceware™ Liquid Nitrogen-Cooled Mini Mortar. A pool of ovaries from six mice was used to reach a minimum weight of 50 mg allowing a chromatin yield of at least 60 μg. Samples were fixed in 10 ml of Fixation Buffer (1.5% methanol-free formaldehyde in 1% PBS) on a roller at room temperature for 15 min. Fixation was stopped by the addition of Stop buffer (Active Motif) for 5 min. Chromatin preparation was continued as per the Active Motif protocol except for the sonication steps that were performed in a Bioruptor® Plus sonication device (Diagenode) with the following settings: 40 cycles of 30 sec ON/30 sec OFF, power "high", constant temperature of 4 °C. ChIP-SICAP experiments were performed in parallel using two biological replicates. ChIP-SICAP allows the identification of chromatin-bound proteins that colocalize with a bait protein (FOXL2 in this study) on DNA. A total of 30 μg of sonicated chromatin was used for each IP using 5 μl of FOXL2 antibody or no-antibody as the negative control. ChIP-SICAP was performed as described by[35]. Briefly after IP, protein complexes were captured on protein G Dynabeads, DNA was biotinylated by Klenow 3' exo in the presence of biotin-7-dATP, followed by TdT in the presence of biotin-11-ddUTP and biotin-11-dCTP, and eluted. Protein-DNA complexes were captured with protease-resistant streptavidin beads[85], formaldehyde cross-linking was reversed, and proteins digested with 300 ng of LysC overnight, followed by 8 h of incubation with 200 ng trypsin. Digested peptides were cleaned using the stage-tipping technique as follows: digested samples were acidified by addition of 2 ul of TFA 10%. For each sample, 50 μl of 80% acetonitrile/0.1% formic acid was aliquoted in a LC-MS Certified Clear Glass 12 × 32 mm Screw Neck Total Recovery Vial (Waters). ZipTip with 0.6 μL C18 resin was pretreated by pipetting 100% acetonitrile twice by aspirating, discarding and repeating. The ZipTip was equilibrated by pipetting 0.1% TFA, three times by aspirating, discarding and repeating. Acidified samples were then pipetted 10 times with ZipTip to allow peptides to bind to the polymer within the tip. Following tip wash in 0.1% TFA, peptides were eluted in 80% acetonitrile/0.1% formic acid in the glass vial and the eluent dried in a speed vac. Peptides were reconstituted in 8 μl of 2% DMSO/0.1% formic acid.

Then, peptides were separated on a 50 cm, 75 μm I.D. Pepmap column over a 70 min gradient to be injected into the mass spectrometer (Orbitrap Fusion Lumos) according to the universal Thermo Scientific HCD-IT method. The instrument ran in data-dependent acquisition mode with the most abundant peptides selected for MS/MS by HCD fragmentation and MS/MS by IT. The raw data were analysed using Proteome Discoverer 2.1 (Thermo Scientific). Briefly, Sequest HT node was used to search the spectra using the UniProt *Mus musculus* database. The search parameters were as follows: The precursor mass tolerance was 10 ppm, and MS/MS tolerance was 0.05 Da. Variable modification included Methionine oxidation and N-terminal acetylation. Fixed modification included Cysteine carbamidomethylation. Trypsin and LysC were chosen as the enzymes, and maximum 2 missed cleavages were allowed. Perculator node was used to eliminate false identifications (FDR < 0.01). Protein ratios in Foxl2 assays ($n = 2$) over no-antibody assays ($n = 2$) were calculated in Proteome Discoverer. The quantification values were exported to R studio to analyse significantly enriched proteins using $t$ test limma package to determine Bayesian moderated t-test p-values and Benjamini–Hochberg (BH) adjusted $p$ values. We considered proteins with mean fold enrichment >2 and adj. $p$ value <0.1 as enriched proteins.

### RNA isolation, RT-qPCR and RNA-seq analysis

RNA was extracted from ovaries using TRIZOL (Thermo Fisher Scientific) and processed as described previously[8]. RNA quality was controlled with the Agilent 2100 Bioanalyzer system. RT-qPCR was performed as previously described[8] using the primers listed below and 18 S as the reference gene for data normalisation. All the statistical analyses were performed using the GraphPad Prism v9 software.

Primers for RT-qPCR (*Gene*: Forward Primer; Reverse Primer), hybridisation temperature = 60 °C:

*FoxL2*: CGGGGTTCCTCAACAACTC; CATCTGGCAGGAGGCGTA

*Wnt4*: ACTGGACTCCCTCCCTGTCT; TGCCCTTGTCACTGCAAA

*Rspo1*: CGACATGAACAAATGCATCA; CTCCTGACACTTGGTGCAGA

*Esr2*: CCATGATTCTCCTCAACTCCA; TGTCAGCTTCCGGCTACTCT

*Amh*: GGGGAGACTGGAGAACAGC; AGAGCTCGGGCTCCCATA

*Cyp19a1*: GAGAGTTCATGAGAGTCTGGATCA; CATGGAACATGCTTGAGGACT

*Sox8*: GACCCTAGGCAAGCTGTGG; CTGCACACGGAGCCTCTC

*Sox9*: TCGGACACGGAGAACACC; GCACACGGGGAACTTATCTT

*Dmrt1*: AAGGCCCCTCCTACTCAGAA; GCTGGAGAGGGAGACCAAG

*Gata1*: TGGGGACCTCAGAACCCTTG; GGCTGCATTTGGGGAAGTG

*Dhh*: CACGTATCGGTCAAAGCTGA; GTAGTTCCCTCAGCCCCTTC

*Ptgds*: GGCTCCTGGACACTACACCT; ATAGTTGGCCTCCACCACTG

*Hsd3b1*: GACCAGAAACCAAGGAGGAA; GCACTGGGCATCCAGAAT

*Cyp17a1*: CATCCCACACAAGGCTAACA; CAGTGCCCAGAGATTGATGA

*Cyp11a1*: AAGTATGGCCCCATTTACAGG; TGGGGTCCACGATGTAAACT

*StAR*: TTGGGCATACTCAACAACC; ACTTCGTCCCCGTTCTCC

*Srd5a1*: CATCTACAGGATCCCACAAGG; TCAATAATCTCGCCCAGGAA

*Trim28*: ATCAGCTGGCTACCGACTCT; GCACGAATCAAGGTCAGGTC

*18S*: GATCCATTGGAGGGCAAGTCT; CCAAGATCCAACTACGAGCTTT

RNA-seq libraries were generated from 600 ng of total RNA using the TruSeq Stranded mRNA LT Sample Preparation Kit (Illumina, San Diego, CA), according to the manufacturer's instructions. Briefly, following purification with oligo d(T) magnetic beads, mRNA was fragmented using divalent cations at 94 °C for 2 min. Cleaved RNA fragments were copied into first-strand cDNA using reverse transcriptase and random primers. Strand specificity was achieved by replacing dTTP with dUTP during the second-strand cDNA synthesis using DNA Polymerase I and RNase H. After the addition of a single 'A' base and adapter ligation to double-stranded cDNA fragments, products were purified and enriched by PCR (30 sec at 98 °C; [10 sec at 98 °C, 30 sec at 60 °C, 30 sec at 72 °C] × 12 cycles; 5 min at 72 °C) to create the cDNA library. Surplus PCR primers were removed by purification with AMPure XP beads (Beckman-Coulter, Villepinte, France), and the quality and quantity of the final cDNA libraries were checked by capillary electrophoresis. Libraries were then sequenced on an Illumina HiSeq 4000 system using single-end 1 × 50 bp, following the Illumina recommendations. Image analysis and base calling were performed with RTA 2.7.3 and CASAVA 2.17.1.1. Reads were mapped to the mm10 assembly of the *Mus musculus* genome using STAR[86] version 2.5.3a. Gene expression quantification was performed from uniquely aligned reads using htseq-count[87] version 0.6.1p1, with annotations from Ensembl version 92 and "union" mode. Comparison between wild-type and cKO samples was performed using the Wald test for differential expression proposed by Love et al.[81] and implemented in the Bioconductor package DESeq2 version 1.16.1.

### Testis and ovary dissociation for single-cell RNA-seq analysis

The testes from an 8-week-old control male were collected and enzymatically dissociated with a modified protocol from[88]. Briefly, tunica

albuginea was delicately removed, and testes were incubated in DMEM (11885084; Gibco) supplemented with 1 mg/mL collagenase (C0130; Sigma-Aldrich), 1 mg/mL hyaluronidase (H3506; Sigma-Aldrich), and 0.8 mg/mL DNase I (dN25; Sigma-Aldrich) at 35 °C for 10 min with gentle agitation. Tissue was centrifuged and incubated in DMEM (11885084; Gibco) supplemented with 1 mg/mL collagenase (C0130; Sigma-Aldrich), 0.025% trypsin-EDTA (25300-054; Gibco), and 0.8 mg/mL DNase I (dN25; Sigma-Aldrich) at 35 °C for 25 min with gentle agitation. Cells were filtered through a 100 μm cell strainer and pre-stained with 100 μg Hoechst dye (B2261; Sigma-Aldrich) and 400 μg DNase I (dN25; Sigma-Aldrich) at 35 °C for 20 min with gentle agitation. Trypsin was quenched with 600 μL fetal bovine serum (F2442; Sigma-Aldrich). Cells were stained with 6 μg Hoechst dye per million cells (B2261; Sigma-Aldrich) and 40 μg DNase I (dN25; Sigma-Aldrich) at 35 °C for 25 min with gentle agitation. Cells were centrifuged, resuspended in DMEM (11885084; Gibco) supplemented with 2% fetal bovine serum (F2442; Sigma-Aldrich) and filtered through a 70 μm cell strainer. Single cells were collected on a BD FACS Aria II by excluding debris (side scatter versus forward scatter), doublets (area versus width) and haploid cells with low DNA content (low Hoechst fluorescence intensity)[88]. FACS-sorting was performed at the Flow Cytometry Facility of the University of Geneva. Cells were centrifuged, resuspended in DMEM (11885084; Gibco) supplemented with 2% fetal bovine serum (F2442; Sigma-Aldrich), counted and immediately processed with a 10X Chromium controller.

Ovaries from two 8-week-old *Trim28^flox/flox*; *Nr5a1:Cre* females and two 8-week-old control females were collected, minced into small pieces and enzymatically dissociated in PBS (10010-015; Gibco) supplemented with 0.25 mg/mL collagenase (C0130; Sigma-Aldrich) and 0.0005% trypsin-EDTA (25300-054; Gibco) at 37 °C for 30 min with gentle agitation. Trypsin was quenched with 200 μL PBS (10010-015; Gibco) supplemented with 3% BSA. Cells were centrifuged, resuspended in PBS (10010-015; Gibco) supplemented with 3% BSA and filtered through a 70 μm cell strainer. Single cells were collected on a BD FACS Aria II by excluding debris (side scatter versus forward scatter) and doublets (area versus width). FACS-sorting was performed at the Flow Cytometry Facility of the University of Geneva. Cells were centrifuged, resuspended in DMEM (11885084; Gibco) supplemented with 2% fetal bovine serum (F2442; Sigma-Aldrich), counted, and immediately processed with a 10X Chromium controller.

### Single-cell RNA-seq library preparation, sequencing, and transcriptomic analyses

Single-cell sequencing was performed on the 10X Genomic platform using 8-week-old *Trim28^cKO* (*Trim28^flox/flox*; *Nr5a1:Cre* mouse line) (2 biological replicates) and control ovaries (2 biological replicates) and testes (1 biological replicate). To increase the numbers of testicular somatic cells, public 10X Genomic data from seven adult mice testes were incorporated[26] (Sup. Tab S2).

After converting the base calls to FASTQ format, reads were processed with the CellRanger v3 count module. This aligned reads with STAR[86] to the GRCm38 reference genome using the M15 (Ensembl 90) GENCODE annotation, and then derived the gene vs. cell expression counts matrices. These pre-processing steps were performed on the Baobab HPC cluster at the University of Geneva. The quality control statistics were checked, such as number of reads per cell and percentage of reads mapped to the reference transcriptome (Data S2). Using Scanpy with Anndata[89], matrices were concatenated across all samples, and cells with <50 genes expressed were removed (and genes expressed in <3 cells). This removed 1,683 of the 57,063 cells. Expression counts were normalised and log-transformed, and highly variably expressed genes were identified. The top 50 PCA components were embedded in the neighbourhood graph with batch correction via BBKNN[90] with the neighbours_within_batch=5 parameter that removed batch effects among

biological replicates (Supplementary Fig. 6A, bottom right). Data were visualised in the transcriptional space with 2D uniform manifold approximation and projection (UMAP)[91] and force-directed graphs (FDG) using the ForceAtlas2 implementation[92], mainly in Jupyter notebooks. Cell clusters were defined with the Leiden algorithm[93] and were annotated by marker gene analyses (Supplementary Fig. 6b). The supporting cell lineage was extracted, represented by Leiden clusters 2 (granulosa/mutant) and 35 (Sertoli). Cluster 2 expressed the granulosa cell markers *serpine2*, *Hsd17b1*, *Inhba*, *Amh*, *Foxl2*, *Nr5a2*, *Fst*, *Wnt4*, *Esr2*. Cluster 35 expressed the Sertoli cell markers *Cldn11*, *Sox9*, *Sox8*, *Rhox8*, *Ptgds*, *Gata1*, *NrOb1*, *Abtb2*. Then, the Leiden clustering was repeated at a higher resolution to separate the granulosa, intermediate, and Sertoli cell populations in three distinct populations that were compared by partition-based graph abstraction (Fig. 2a, right)[94]. These cells were ordered along a diffusion pseudo-time[30], setting the starting root cell in the granulosa control cells, and were plotted according to the expression of the differentially expressed genes calculated with Wilcoxon rank-sum tests (with Benjamini-Hochberg multiple testing corrections) among the three clusters, filtering genes that had a fold-change <1.5 and genes that were expressed in ≥0.8 proportion of the cluster in which they were downregulated. The top 70 upregulated genes were plotted for each cluster in Fig. 2d, and the full sets of filtered upregulated genes are shown in Sup. Tab S3. The mean expression across marker genes was also plotted using previously defined modules: F23 for granulosa, F17 for Sertoli, and F1-F10 for progenitor cell markers; see Fig. 5 in ref. 29. The functional annotations that were enriched among gene lists were examined using hypergeometic tests performed via g:Profiler[95]. To calculate the enrichments in Sup. Data S4, 70 upregulated genes were used in the intermediate cell population as the foreground gene set, and all the mouse genes expressed in the concatenated Anndata matrix after pre-processing as the background gene set.

### Antibodies

Polyclonal antibodies against SUMO1 and SUMO2/3 were produced by injecting rabbits with recombinant His-tagged mouse SUMO1 and SUMO-3 proteins produced and purified from bacteria as previously described[96]. Briefly, His-SUMO expressing plasmids (pTE1-E2-HisSU1 and pTE1-E2-HisSU3) were transfected in bacteria BL21/DE3 gold (Stratagene) and induced by 1 mM IPTG for 6 h at 25 °C after reaching 0.4 OD. After lysis, bacterial extracts were purified with Ni-NTA resin (Qiagen) following manufacturer instructions, with 250 mM imidazole in elution buffer. SUMOs containing fraction were then subjected to a second purification step using Superdex 75 gel filtration column (GE Helthcare). Rabbit sera were affinity-purified using GST-SUMO1 or GST-SUMO2 coupled to CnBr-activated Sepharose (SIGMA) using manufacturer instructions. Their specificity was confirmed by immunoblotting using recombinant mouse SUMO1 and SUMO2. Dilution for immunofluorescence: 1/300 For ChIP-seq: 2 μg /IP.

Other antibodies:

FOXL2 (rabbit)[97]. Immunofluorescence: 1/400. ChIP-seq and ChIP-SICAB: 5 μl (2 μg) / IP.

SOX9 (rabbit)[98]. Immunofluorescence: 1/400.

SOX8 (guinea pig)[99]. Immunofluorescence: 1/300.

DMRT1 (rabbit)[100]. Immunofluorescence: 1/400.

TRIM28 (mouse)[101]. Immunofluorescence: 1/1000.

TRIM28 (rabbit)[102]. ChIP-seq: 2 μg / IP.

V5-HRP: Invitrogen. Ref P/F 46-0708. Western blotting:1/5000

HA-HRP:. Invitrogen. Ref 26183-HRP. Western blotting: 1/5000

V5: Invitrogen. Ref 37–7500. Western blotting: 1/2000.

HA: Invitrogen. Ref MA1-12429. Western blotting: 1/2000.

FLAG: Invitrogen. Ref MA1-91878. Western blotting: 1/3000

Tubulin: Sigma. Ref T9026. Western blotting: 1/3000

## Histology, immunofluorescence, and confocal microscopy

Postnatal and adult ovaries were collected, fixed in 4% paraformalde-hyde, and paraffin-embedded. Then, 4-µm sections were cut and processed for histology and immunofluorescence. Sections were stained with periodic acid-Schiff (PAS) stain using standard protocols. Immuno-fluorescence was performed as previously described[103]. Overnight incubation with primary antibodies at 4 °C was followed by incubation with the appropriate secondary antibodies (1/800) (Alexa-Ig, Mole-cular Probe). Nuclei were stained with DAPI (Sigma). Images were captured with a Zeiss Axioimager Apotome microscope. For quantifi-cation of SUMO1 and SUMO2 immunofluorescence signals, images were captured with a Zeiss LSM780 confocal microscope and analysed with the Imaris V9.5. Statistical analyses were performed using the GraphPad Prism v9 software using two-way ANOVA with Dunnett's multiple comparisons test.

## Plasmids, cell culture and SUMOylation assays

The Flag-TRIM28 expression plasmid was described previously[101]. The TRIM28 C651F mutation was generated using the Quick-change II site-directed PCR mutagenesis kit (Agilent) and the sequence was verified by Sanger-sequencing. The pcDNA3-HA-SUMO2 expression plasmid was obtained from Dr. Ronald Hays. The pcDNA3.1-FOXL2-V5 (V5-tag-ged in C-terminal) expression plasmid was obtained from Prof Reiner Veitia. To generate the pcDNA3.1-RUNX1-V5 construct (i.e., RUNX1 C-terminally tagged with V5), the human RUNX1 open reading frame was PCR-amplified from the RUNX1-pCSdest plasmid (a gift from Roger Reeves. Addgene plasmid # 53802; http://n2t.net/addgene:53802; RRID: Addgene_53802) and subcloned in the pcDNA3.1/V5-HIS TOPO vector (Invitrogen) following the manufacturer's instructions. The sequence was verified by Sanger-sequencing.

HEK293T cells were obtained from ATCC (ref CRL-1573). For SUMOylation assays in transfected cells: HEK293T cell culture, trans-fection, and SUMOylated FOXL2 and RUNX1 immunoprecipitation conditions were as described previously[51]. FOXL2-V5 and RUNX1-V5 were immunoprecipitated using V5 agarose affinity gel (SIGMA ref: A7345). IP complexes were analysed by western blotting using anti-V5 and anti-HA-HRP-coupled antibodies (see Antibodies section). For inputs, crude cellular extracts were probed with non-coupled anti-bodies (see Antibodies section).

## Steroid quantification

Steroids (testosterone, androstenedione, β-oestradiol and estrone) were quantified in gonads from 4-month-old control females, $Trim28^{cKO}$ females, and control males ($n = 4$/group) by liquid chro-matography coupled to tandem mass spectrometry (LC-MS/MS). Briefly, adult tissues were homogenised in methanol containing mixed internal standards. After evaporation, dried residues were resuspended in 600 µl of methanol, loaded onto an Oasis Prime HLB (Waters) cartridge, and sequentially washed with 0.1% formic acid in 35% methanol, and 0.1% ammonia solution in 35% methanol. Sam-ples were eluted with 45 ml of methanol and then diluted with 55 µl of distilled water. After mixing, 30 µl of each sample was injected for Ultra High-Performance Liquid Chromatography (UPLC) analysis. LC-MS/MS analysis was performed on a UPLC Acquity system (Waters Corporation) coupled to a triple-quadrupole XevoTQD mass spectrometer (Waters Corporation). Steroids were analysed with a reversed-phase column (Acquity UPLC HSS T3 column C18, 1.8 µm, 2.1 × 50 mm, Waters Corporation). The chromatographic mobile phase was 2 mM ammonium acetate with 0.1% formic acid (v/v) in water (Phase A) and 2 mM ammonium acetate with 0.1% formic acid (v/v) in methanol (Phase B) delivered at a flow rate of 0.6 mL/min at 50 °C. The gradient was 0–1 min, 55% A; 1–3.5 min, 50% to 35% A; 3.5–3.51 min, 35% to 5% A. To equilibrate the column again, at 3.51 min the gradient remained in the initial conditions for 1.49 min. The total run time was 5 min. Steroid ionisation was performed using positive electrospray ionisation of ($[M + H]^+$) with the following settings: capillary voltage, 500 V; cone voltage, 35 V; desolvation gas, 1000 L/h; cone gas, 80 L/h; desolvation tempera-ture, 500 °C; and source temperature, 150 °C[104].

## Statistics and reproducibility

Statistical analyses for next-generation sequencing data are described in RNA-seq, scRNA-seq ChIP-seq methods section. The other statistical analyses were performed with GraphPad Prism 9 software. Details of individual tests are outlined within each figure legend, including a number of replications performed ($n$) and the reported error as the standard error of the mean (SEM) for every figure except for supple-mentary figure 7b where error bars are the mean ± SD. At least three independent biological replicates were analysed with at least three technical replicates. Statistics were calculated with two-way ANOVA with Dunnett's multiple comparisons test (to compare every mean with every other mean), ordinary one-way ANOVA with Tukey's mul-tiple comparisons test (to compare every mean to a control mean). For histological analysis, at least three independent biological replicates from independent mating, if possible, were analysed and images are representatives of all replicates. No statistical method was used to predetermine sample size, samples were allocated into the experi-mental groups by genotype and no data were excluded from the analyses. For the biological experiments, investigators were not blin-ded to group allocation for data collection and analysis since the same investigator designed and performed the experiments.

## Reporting summary

Further information on research design is available in the Nature Research Reporting Summary linked to this article.

## Data availability

All data are available in the main text or supplementary materials. RNA-seq and ChIP-seq data have been deposited in the Gene Expression Omnibus under accession number: -GSE166385 (RNA-seq and ChIP-seq)-GSE166444 (scRNA-seq). The mass spectrometry pro-teomic data have been deposited in the ProteomeXchange Con-sortium via the PRIDE partner repository with the dataset identifier PXD024439. Source data are provided with this paper.

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

## Acknowledgements

This study was supported by the Centre National de la Recherche Scientifique (CNRS), the University of Montpellier, and by a grant from 'Agence Nationale de la Recherche' (ANR-16-CE14-0020-SexMaintain to F.P.). R.M. and R.L.B. are funded by the Francis Crick Institute. The Francis Crick Institute receives its core funding from Cancer Research UK (FC001107), the UK Medical Research Council (FC001107), the Wellcome (FC001107), and the UK Medical Research Council (U117512772). We thank Nitzan Gonen from Bar-Ilan University for scientific discussions and critical reading of the manuscript. We are grateful to Monsef Benkirane and Dominique Giorgi for their continual support of the project. We thank the staff of the 'Reseau d'Histologie de Montpellier'(RHEM) for paraffin embedding of tissue samples and histology staining. We thank Marie-Pierre Blanchard, and Amelie Sarrazin of the Montpellier Imaging Facility (MRI) for their help with microscopy experiments. We thank the staff of the RAM-CECEMA animal care facility (University of Montpellier). We thank Francoise Kuhne for her technical assistance in single-cell analysis. We thank the GenomEast platform, particularly Violaine Alunni for the preparation of RNA-seq libraries and Romain Kaiser for sequencing. We thank Professor Michael Wegner and Prof David Zarkower for their generous gift of the anti-SOX8 and anti-Dmrt1 antibodies respectively. We thank Professor Reiner Veitia for the generous gift of the pcDNA3.1-FOXL2-V5 expression plasmid. We thank the Freiburg Galaxy server that was used for some calculations. We thank Elisabetta Andermarcher for manuscript editing.

## Author contributions

M.R. and F.P. designed the study. M.R, S.D., L.L. and B.B.B. performed histological and RT-qPCR experiments. S.N. and F.P. designed the scRNA-seq. C.R., M.R., Y.N. and S.N. performed and analysed scRNA-seq. F.P. designed and performed ChIP-seq experiments. All bioinformatics analyses, except for scRNA-seq, were performed by S.L. F.C. designed and created the mouse line carrying Trim28[Phd] allele. A.P. and A.L.N. performed steroids dosage. G.B. and D.W. produced and purified the SUMOs and FOXL2 antibodies, respectively. R.L.B., R.M. and M.R.R. designed, performed, and analysed FOXL2 ChIP-SICAP. F.P. wrote the manuscript. All authors reviewed and added input to the manuscript.

## Competing interests

The authors declare no competing interests.
