## [Peer Review File · Nature Communications]

TRIM28-dependent SUMOylation protects the adult ovary from activation of the testicular pathwayREVIEWER COMMENTS

Reviewer #1 (Remarks to the Author):

TRIM28-dependent SUMOylation protects the adult ovary from activation of the male pathway

The authors found that upon loss of Trim28, ovarian granulosa cells transdifferentiate to Sertoli cells through an intermediate cell type, different from gonadal embryonic progenitors. They show that TRIM28 binds in the proximity of FOXL2 on chromatin, which correlates with the maintenance of the female pathway and repress male-specific genes. The E3 SUMO ligase activity of Trim28 is at least in part involved in somewhat differential occupancy of SUMOs on the chromatin of female-specific genes compared to male ones. The manuscript contains potentially a very interesting news. However, clarification of the following points would strengthen the manuscript.

Specific comments:

The authors state that the TRIM28-dependent SUMOylation, which represents less than 10% of the whole SUMOylation landscape, is sufficient to prevent adult sex-reversal. They also state that TRIM28 induces relatively sharp peaks of SUMOylation on chromatin (<1 kb). Overall, a lot of SUMO “peaks” (~ 250,000) was found in this work. This large number probably reflects the parameters and programs used for peak detection. The reviewer found the “SUMO peaks” often more like chromatin regions with relatively low signal to noise ratio in peak detection. MACS2 program has been used for the TRIM28 and FOSXL2 ChIP-seq data, but Peak caller program for the SUMO ChIP-seq data. For Peak caller, no specific parameters have been given. The reasoning for the usage of different programs is lacking, but it seems that the authors have been searching for SUMOs regions/broad “peaks” rather than sharp transcription factor peaks. Therefore, the SUMO “peaks” are wide, and often difficult to envision in the track images. Given all this, it is difficult to get a clear idea of the “hypo/hyper”-SUMOylated regions from the Fig. 5 heatmap or the Fig. S14 MA plots. Histograms and/or boxplots would be needed to facilitate the comparisons and the effects of Trim28 defects.

As the authors also mention that many hypo-SUMOylated “peaks” did not overlap with TRIM28 or FOXL2, indicating that other transcription factors or chromatin-associated proteins display TRIM28-dependent SUMOylation, it would be pertinent to use SUMO proteomics to identify these proteins. Moreover, experiments combining single-cell ATAC-seq and RNA-seq should answer the question whether in the absence of functional TRIM28, FOXL2 indeed loses its capacity to repress the male pathway.

Reviewer #2 (Remarks to the Author):

The manuscript by Rossitto et al reports the latest example of an XX gonad masculinisation phenotype in the postnatal mouse ovary following gene deletion. Whilst this phenotype has previously been associated with transcription factor (TF) deletion – most notably FOXL2 – here the cause of sex reversal is loss of a protein involved in SUMOylation, TRIM28. Using some nice genetic approaches, the authors show that this SUMOylation activity is required

to maintain ovarian somatic cell identity. They also use genome-wide approaches to show that SUMOylation appears to function at ovarian genes to stabilise their activity through impacts on FOXL2 and other key TFs. This study strengthens our understanding of the global role played by sex-determining genes in crafting an ovarian or testicular chromatin architecture and will be of interest to many in the field of sex determination, chromatin, post-translational modification and maintenance of cell fate more broadly. But the manuscript lacks some clarity in places and the comments below are aimed at trying to improve its clarity and impact.

Title: The title refers to the 'male' pathway. Given that women with complete androgen insensitivity syndrome have testes, and given how the classification of sex and gender is increasingly controversial, it is probably best not to identify sex with gonadal sex. The 'testicular pathway' or 'Sertoli cell pathway' would be preferable – and this is the case throughout the whole manuscript, where 'male' and 'female' are used to denote 'testis' and 'ovary'.

Abstract: TRIM28 is said to 'bind in close proximity' – but how does an enzyme bind chromatin? It should be made clear early in the manuscript what sort of protein TRIM28 is (not DNA binding) and how recruitment to chromatin is presumably via proteins that can ultimately bind DNA.

Also, why is it 'remarkable' that an E3 SUMO ligase functions to regulate SUMOylation?

Page 2, line 15: TRIM28 is a partner of testicular SOX9 – so isn't unexpected that it functions to maintain the ovarian soma? Why is this 'paradox' not discussed? Do the SUMO peaks identified later in the manuscript coincide with SOX9 peaks following ChIPseq?

Page 2: lines 21-23 state that TRIM28 protein is not detected in cko gonads – is this at birth? How early is this loss seen? Can it be inferred that TRIM28 is not required for primary sex determination unless it is shown that TRIM28 is absent in fetal gonads during the sex determining stages? Perhaps loss is gradual and this contributes to a late onset phenotype?

P4, line 10: perhaps a little more explanation would be useful (especially for non-sex expert readers) why the medullary-cortical polarity in the appearance of sex reversal is explained by what is known about the wave-like appearance of granulosa cells during fetal development.

P4: the use of 'transdifferentiation' and 'reprogramming' is first used here. Here (or in the conclusion section) it should be summarised briefly why the best explanation for the sex reversal is reprogramming/transdifferentiation, as opposed to, for example, loss of granulosa cells and the appearance of Sertoli-like cells from another (previously unrecognised) progenitor. The paper by Uhlenhaut et al (2009 Cell) that first reported this phenotype following Foxl2 ablation went to great lengths to exclude other explanations. Did the authors, for example, assess apoptosis in the cko gonads? Could loss of germ cells contribute to the sex reversal in this line? Why is transdifferentiation the only possible explanation?

P6: What is the explanation for why Rspo1 expression appears to be unaffected? RSPO1 is a major pro-WNT factor, so why is WNT disrupted in the cko?

P7: the term 'pseudo-Sertoli cell' is new and should be explained in further detail. Is it the same as 'Sertoli-like'?

P8: the distribution of cko cells shown in Fig 2a shows a few 'intermediate' cells, but the bulk

of cko cells appear to be (spatially, so transcriptionally) closely associated with control ovarian somatic cells i.e. they form part of a larger ovarian cluster. So, the distribution still appears bi-modal whatever the genotype, apart from a small number of cells between ovarian and testicular control clusters. Is this expected for this sex reversal phenotype? Are there no Sertoli cells in the cko gonad? What are the cko granulosa cells transdifferentiating into? Some further clarification is important here. Just how sex reversed are these sex-reversed cko gonads?

P13: Do the PHD/PHD homozygotes die early, like the straight KO? This should be made clear.

P15: Were any of the SUMO1/2 peaks occupied by RUNX1? Given the Nicol et al (Nat Commun 2019) report of overlap of FOXL2 and RUNX1 chromatin occupancy, might TRIM28 SUMOylate both FOXL2 and RUNX1? Was Cldn11 a site of hyper-SUMOylation?

P21: the manuscript ends with a comment about human fertility. But perhaps more interesting is whether there is any genetic evidence of a role for TRIM28 in human sex determination (like RSPO1)? There are a number of exome sequencing resources that could be examined. Or a role in female POI? Could the authors speculate?

Finally, it would be good for the authors to briefly clarify in the Discussion whether the activation of the testicular pathway is simply a consequence of the down-regulation of the ovarian pathway i.e. due to their mutual antagonism, or whether TRIM28 functions itself to directly oppose the testicular pathway, or both the above.

Reviewer #3 (Remarks to the Author):

The manuscript by Rossitto et al identifies a new role of TRIM28 in the maintenance of the postnatal ovary identity. Conditional KO of Trim28 in mouse ovary leads to ovary-to-testis sex reversal and scRNA-seq demonstrates transdifferentiation of granulosa cells into an intermediary state resembling testis Sertoli cells. The Trim28 cKO phenotype was similar to Foxl2 postnatal KO phenotype. ChIP-seq for TRIM28 in adult ovary revealed overlap with FOXL2 ChIP-seq, suggesting that TRIM28 may act along with FOXL2 or maybe? through posttranslational regulation of FOXL2 to maintain ovarian identity. TRIM28 has indeed an intrinsic E3 SUMO ligase activity. The authors created a point mutation mouse model that abrogates this function, and the phenotype of these mice was similar to that of the Trim28 cKO, indicating that TRIM28 acts through this SUMO ligase function to maintain ovarian identity. Comparison of SUMO1-2/3 ChIP-seq between control ovary and both Trim28 mutant models revealed significant changes in sumoylation in the mutant ovaries, including around granulosa/Sertoli specific genes.

Overall, this is a well-written manuscript, with judicious use of various transcriptomic / genomic techniques and mouse models to characterize in details the functions of TRIM28 in the ovary.

Here are my comments:

My main comment relates to the role TRIM28 and sumoylation in transcriptional regulation. TRIM28 and sumoylation are commonly associated with transcriptional repression. How do you reconcile this with your findings that genes downregulated are hypsumoylated whereas

genes upregulated are hypersumoylated? This seems to go against the consensus of hypsumoylation is associated with transcriptomic activation. This needs to be further discussed.

Since TRIM28 is mostly known as a transcriptional repressor, what is the first event that occurs in the transdifferentiation of cKO ovaries? Upregulation of male genes, or downregulation of female genes? Sox8 transcript levels are already high at 0.5m in Trim28 cKO ovaries (Fig.1C), to the level of control testes. Could de-repression of some Sertoli genes happen before the downregulation of granulosa genes in cKO ovaries? Have you looked at an earlier time-point than 20 days if SOX8 protein expression precedes the loss of FOXL2 protein rather than appearing after FOXL2 loss in supporting cells?

In Fig. 1A, the authors show that SOX8 appears before SOX9. There are however some discrepancies between immunos at 20d and 8w (Fig. S3 bottom). While at 20d there seem to be more SOX8+ cells than SOX9+ cells (Fig 1A), at 8w old, there are more SOX9 single positive cells in follicles than SOX8 single positive cells. How do you explain this difference? Similarly, there are some discrepancies in FOXL2 labelling in 8-week-old ovaries in Fig. S3. In the double immunostaining FOXL2/SOX8, it is very clear that some follicular cells are still expressing FOXL2 in most follicles in the medulla, particularly at the periphery of the follicles. However, in the below double immunostaining FOXL2/SOX9, FOXL2 appears to be gone from most follicles in the medulla. Is there a lot of inter-animal variation in the phenotype? Which phenotype is more common? It is actually quite interesting that the double labeling FOXL2/SOX8 shows that almost all the supporting cells in the one layer at the periphery of the follicle / pseudo-tubular structure maintain FOXL2 expression (Fig S3). Could these cells be more resistant to transdifferentiation?

Why do you think some cKO supporting cells transdifferentiate while others seem to maintain a pretty normal granulosa transcriptome based on scRNA-seq? Is it the fetal origin of the 2 waves of granulosa cells, medulla/cortex that prime them into their different capacity to respond to masculinization signals?

FOXL2 functions, for instance transcriptional repression, and stability rely on its sumoylation state. TRIM28 cKO phenotype is similar to FOXL2 postnatal KO. Do you think TRIM28 acts solely on FOXL2 to maintain ovarian identity or does it play a more global genomic role / targeting other TFs involved in supporting cell identity as well? Have you looked at FOXL2 protein sumoylation state in ovaries of the two TRIM28 mutant models?

The authors mention that heterozygous Nr5a1:Cre;Trim28 flox/+ have no phenotype. Since these mice are used for the cross with the Trim28 C651F mutants to characterize the effects of the sole expression of this mutated Trim28 protein in the ovary, it is relevant to provide the data proving normal ovarian phenotype, such as PAS staining in Fig S13. Have the authors looked at gene expression to confirm that haploinsufficiency of Trim28 in Nr5a1 cells does not affect granulosa/Sertoli genes (even though these mice are fertile), and potentially contributes to the PHD/cKO mouse phenotype?

Fig 5C: The tracks for TRIM28 and FOXL2 ChIP-seq need to be directly added in order to easily compare with all the SUMO tracks. Going back and forth with Fig.3C, which has different track sizes, is not ideal.

Throughout the manuscript, the authors state that granulosa cells are reprogrammed into Sertoli cells. However, the single cell RNA-seq analysis clearly shows that these transdifferentiated cells never reach a full Sertoli cell state as they do not cluster with the normal testis Sertoli cell and rather stay in an intermediary state between granulosa and Sertoli identity. It would therefore be more accurate to call these cells "Sertoli-like" rather than Sertoli.

Reviewer #4 (Remarks to the Author):

This is a well-written manuscript describing a series of experiments establishing the role of TRIM28 in maintaining the female characteristics of the mouse ovary. The authors created a TRIM28 knock-out mouse (cKO) and observed progressive female-to-male transition in tissue structure, cell states, and known male/female markers. At multiple time points after birth the cKO ovaries displayed gradual conversion of the female-specific granulosa cells to a transcriptional state similar to the male-specific Sertoli cells, as shown by the molecular markers for the male and female lineages. Single-cell RNAseq data showed that the transdifferentiation did not produce bona fide Sertoli cells, rather an intermediate cell type between granulosa and Sertoli.

Starting from Figure 3, the study delved into detailed molecular mechanisms, focusing on the interaction between TRIM28 and FOXL2. ChIP-seq peaks for TRIM28 tend to coincide with those for FOXL2 (55%-62% overlap), suggesting that TRIM28 and FOXL2 jointly maintain the female fate, and TRIM29-cKO resulted in the upregulation of a host of male-specific genes. These data are consistent with prior studies of female-male transdifferentiation using other models, such as targeting FOXL2 or ESR1-2 in adult females.

The last part of the study tested the idea that the role of TRIM28 is mediated by SUMOylation. The authors generated a line carrying a point mutation in TRIM28 (called PHD) that removes its SUMO-E3 ligase activity and found that PHD/cKO and cKO ovaries are similarly affected. This result indicates that TRIM28 maintains the female state by its E3-SUMO ligase function. Protein localization/quantification by microscopy and scRNAseq experiments added more details to the comparison of PHD/cKO and cKO.

I have a few minor comments.

1. In Fig. S2 the oocytes positive for TRIM (red) in cKO ovaries are significantly larger than the wild type. It is not clear how to explain that.
2. All the findings are from antral follicles. It's possible that the expression (or lack of it) of TRIM becomes important at the later stages of folliculogenesis. It would be interesting to see if this mutation affects follicular reserve. At what stage of folliculogenesis does stabilization become important?
3. Fig. S4 - deposition "of basal laminae (green arrow)" is better determined with IHC. Is the composition in the mutant ovary different from the control?
4. It would be useful to clearly state how the Theca cells are affected by cKO. In Fig. S6 the insert is too small to see. It seemed that Clusters 23 and 27 are Theca cells. The number of cells by cluster and by condition can be reported.

5. In Fig. 5A and 5B, are the genes shown in the same order across the three panels: Cont, cKO, and PHD? That is, is each row for the same gene?

6. The data in GEO is not accessible. If it is temporarily blocked, there should be a reviewer token for access.

Point-by-point response to the reviewers.

We thank the referees for their careful reading and many thoughtful comments on the original submission. As described in detail below, we tried to address all their concerns through a substantial revision of the text by adding some clarifications, by changing how the results are described in some figures, and by adding some additional experiments.

For each reviewer, in the paragraph “remarks to the authors”, the references highlighted in green correspond to our answers in the respective paragraph “answer to the reviewer”. Referee comments (in black) and authors responses (in blue). The place of the modification asked by each reviewer in the manuscript is highlighted in yellow.

Reviewer #1 (Remarks to the Author):

TRIM28-dependent SUMOylation protects the adult ovary from activation of the male pathway

The authors found that upon loss of Trim28, ovarian granulosa cells transdifferentiate to Sertoli cells through an intermediate cell type, different from gonadal embryonic progenitors. They show that TRIM28 binds in the proximity of FOXL2 on chromatin, which correlates with the maintenance of the female pathway and repress male-specific genes. The E3 SUMO ligase activity of Trim28 is at least in part involved in somewhat differential occupancy of SUMOs on the chromatin of female-specific genes compared to male ones. The manuscript contains potentially a very interesting news. However, clarification of the following points would strengthen the manuscript.

Specific comments:

The authors state that the TRIM28-dependent SUMOylation, which represents less than 10% of the whole SUMOylation landscape, is sufficient to prevent adult sex-reversal. They also state that TRIM28 induces relatively sharp peaks of SUMOylation on chromatin (<1 kb). Overall, a lot of SUMO “peaks” (~ 250,000) was found in this work. This large number probably reflects the parameters and programs used for peak detection. The reviewer found the “SUMO peaks” often more like chromatin regions with relatively low signal to noise ratio in peak detection. MACS2 program has been used for the TRIM28 and FOSXL2 ChIP-seq data, but Peak caller program for the SUMO ChIP-seq data. For Peak caller, no specific parameters have been given (R1_1). The reasoning for the usage of different programs is lacking, but it seems that the authors have been searching for SUMOs regions/broad “peaks” rather than sharp transcription factor peaks. Therefore, the SUMO “peaks” are wide, and often difficult to envision in the track images (R1_2). Given all this, it is difficult to get a clear idea of the “hypo/hyper”-SUMOylated regions from the Fig. 5 heatmap or the Fig. S14 MA plots. Histograms and/or boxplots would be needed to facilitate the comparisons and the effects of Trim28 defects (R1_3).

As the authors also mention that many hypo-SUMOylated “peaks” did not overlap with TRIM28 or FOXL2, indicating that other transcription factors or chromatin-associated proteins display TRIM28-dependent SUMOylation, it would be pertinent to use SUMO proteomics to identify these proteins (R1_4). Moreover, experiments combining single-cell ATAC-seq and RNA-seq should answer the question whether in the absence of functional TRIM28, FOXL2 indeed loses its capacity to repress the male pathway (R1_5).

Responses to reviewer #1

We thank reviewer 1 for the constructive remarks about the manuscript, and here are our answers to the comments.

R1 1 “For Peak caller, no specific parameters have been given.”

Response: Spp v1.15.5 was run through the ENCODE pipeline v1.3.6 using the following parameters: “-npeak=300000 -fdr=0.01”. The fragment length parameter “-speak” was set according to the value estimated by the ENCODE pipeline.

More details about ENCODE recommendations can be found in:

https://docs.google.com/document/d/11G_Rd7fnYgRpSIqrIfuVIAz2dW1VaSQThzk836Db99c/edit.

This has been added in the Methods section (**Rev#1-1**)

R1 2: “The reasoning for the usage of different programs is lacking, but it seems that the authors have been searching for SUMOs regions/broad “peaks” rather than sharp transcription factor peaks. Therefore, the SUMO “peaks” are wide, and often difficult to envision in the track images.”

Response: We chose to use the peak caller that best fits our experimental design and data, and also following the ENCODE consortium recommendations (^{1,2}). (**Rev#1-2**)

SUMO peaks: SUMO peaks were sharp and thus were detected using the ENCODE pipeline v1.3.6 (³) with parameters set for transcription factors (see before).

The pipeline wraps spp ⁴ v1.15.5 and includes selection of **highly reproducible** peaks between biological replicates using the IDR method ⁵. 250,000 peaks represent the total number of peaks for all conditions we analyzed. Here is the number of peaks per condition:

Condition	N. of peaks
WT_SUMO1	164,138
KO_SUMO2	238,470
KO_SUMO1	199,261
PHD_SUMO2	224,923
WT_SUMO2	218,451
PHD_SUMO1	189,847

As the sum of all unique peaks identified across the conditions corresponds to 250,000 peaks, a large proportion of peaks are shared among the different conditions, and some of them showed different intensities among conditions, as we demonstrated in the differential binding analysis.

TRIM28 and FOXL2: We found that TRIM28 and FOXL2 had large peaks with subpeaks within them. We decided to run the peak calling to detect peaks as the largest significant enrichment and thus, used a peak caller designed and tuned accordingly. We used MACS2 because it is an efficient (^{6,7}) and widely used (^{8,9,10}) peak caller and is an alternative to spp in the ENCODE ChIP-seq pipeline (¹¹). We used it out of the ENCODE pipeline to finely tune the parameters. As we compared FOXL2 and TRIM28 results, we used the same tools to detect peaks in experiments on these two factors.

1. Landt SG, Marinov GK, Kundaje A, et al. ChIP-seq guidelines and practices of the ENCODE and modENCODE consortia. *Genome Res.* 2012;22(9):1813-1831. doi:10.1101/gr.136184.111

2. Data Processing Pipelines – ENCODE. Accessed September 14, 2021. <https://www.encodeproject.org/pipelines/>
3. Release v1.3.6 · ENCODE-DCC/chip-seq-pipeline2. GitHub. Accessed September 14, 2021. <https://github.com/ENCODE-DCC/chip-seq-pipeline2/releases/tag/v1.3.6>
4. Kharchenko PV, Tolstorukov MY, Park PJ. Design and analysis of ChIP-seq experiments for DNA-binding proteins. *Nat Biotechnol.* 2008;26(12):1351-1359. doi:10.1038/nbt.1508
5. Li Q, Brown JB, Huang H, Bickel PJ. Measuring reproducibility of high-throughput experiments. *Ann Appl Stat.* 2011;5(3):1752-1779. doi:10.1214/11-AOAS466
6. Zhang Y, Liu T, Meyer CA, et al. Model-based Analysis of ChIP-Seq (MACS). *Genome Biol.* 2008;9(9):R137. doi:10.1186/gb-2008-9-9-r137
7. Thomas R, Thomas S, Holloway AK, Pollard KS. Features that define the best ChIP-seq peak calling algorithms. *Brief Bioinform.* 2017;18(3):441-450. doi:10.1093/bib/bbw035
8. Wu HJ, Landshammer A, Stamenova EK, et al. Topological isolation of developmental regulators in mammalian genomes. *Nat Commun.* 2021;12:4897. doi:10.1038/s41467-021-24951-7
9. Xu Q, Georgiou G, Frölich S, et al. ANANSE: an enhancer network-based computational approach for predicting key transcription factors in cell fate determination. *Nucleic Acids Res.* 2021;49(14):7966-7985. doi:10.1093/nar/gkab598
10. Bag I, Chen S, Rosin LF, et al. M1BP cooperates with CP190 to activate transcription at TAD borders and promote chromatin insulator activity. *Nat Commun.* 2021;12:4170. doi:10.1038/s41467-021-24407-y
11. Transcription factor ChIP-seq 2 (unreplicated) – ENCODE. Accessed September 14, 2021. <https://www.encodeproject.org/pipelines/ENCPL481MLO/>

R1 3: “Given all this, it is difficult to get a clear idea of the “hypo/hyper”-SUMOylated regions from the Fig. 5 heatmap or the Fig. S14 MA plots. Histograms and/or boxplots would be needed to facilitate the comparisons and the effects of Trim28 defects.”

Response: We thank reviewer #1 for this suggestion. To answer this comment and to improve the understanding of the results, we showed the data from heatmaps in Figure 5 as boxplots in Fig 5A where the SUMO reads are quantified (with a log₁₀ scale) in control, cKO and PHD mutants. Using the same approach, we quantified the number of TRIM28 reads in control at the deregulated SUMO peaks (Fig 5B). We provide in the manuscript a new version of Figure 5.

R1 4: “As the authors also mention that many hypo-SUMOylated “peaks” did not overlap with TRIM28 or FOXL2, indicating that other transcription factors or chromatin-associated proteins display TRIM28-dependent SUMOylation, it would be pertinent to use SUMO proteomics to identify these proteins.”

Response: This is an interesting suggestion that we effectively plan to address in a future study. Analyzing the SUMOylome in a whole organ is already on its own an important project and a difficult task because less than 1% of each protein appears to be SUMOylated, particularly for transcription factors. Knowing that SUMOylation is extremely transient and unstable, it would

be risky to dissociate and purify the different cell populations from control and mutant ovaries to perform a SUMOylome analysis using classical approaches (Barysh et al 2014, PMID: 24651501).

Therefore, specific genetic tools are required, such as the CAG-SUMO mouse line (Yang et al 2014, PMID: 24569813) where conditional expression of tagged SUMO1/2/3 can be induced in a cell population via Cre activation. We plan to import this mouse line in our laboratory to identify transcription factors that are de-novo SUMOylated in the cKO and PHD mutants. Several other transcription factors are known to maintain the granulosa cells fate, such as ESR1 and 2 (Couse et al 1999, PMID 10600740) and RUNX1 (Nicol et al 2019, PMID 31712577), and they also might be regulated by SUMOylation. Therefore, to provide some additional data to address reviewer 1's comment and improve the manuscript we performed two additional experiments:

-ESR2 might be a good candidate because its stability depends on SUMOylation (Picard et al 2012, PMID: 22586270). It is the major oestrogen receptor expressed in granulosa cells and it is also required for ovarian maintenance (Couse et al 1999, PMID: 10600740). We tested the overlap between hypo-SUMOylated peaks and ESR2, using the ChIP-seq dataset for ESR2 in adult ovary recently published by David Zarkower's laboratory. We found that hypo-SUMOylated peaks coincided with ESR2 peaks, but to a lower extent than what observed for FOXL2. Although, it would have been better to compare ESR2 and FOXL2 ChIP-seq data obtained at the same time, David Zarkower's laboratory datasets are very robust. Therefore, this suggest that ESR2 is a good candidate. See Rev#1-3 in the manuscript, and a modified version of Fig S15 (now Fig. S17).

-We performed in vitro SUMOylation using cell transfection experiments and found that FOXL2 is SUMOylated by WT TRIM28, but not by the C651F mutant. Again, this strongly supports the hypothesis that TRIM28 SUMOylates FOXL2. Concerning RUNX1, it was difficult to look for overlapping peaks in the RUNX1 ChIP-seq dataset published by Nicol et al in 2019. Indeed, the experiments were performed using foetal gonads: i) foetal ovary is not the same as adult ovary and ii) due to the small amount of starting materials these experiments provided a smaller number of peaks compared with experiments done using adult tissue. Therefore, we also tested in vitro SUMOylation of RUNX1. As observed for FOXL2, we saw a specific SUMOylation in the presence of TRIM28. See the new Figure S18, and Rev#1-4 in the manuscript.

We provided the results on ESR2 in a modified version of Fig S17 (previously named Fig S15 in the first version of the manuscript) and those on FOXL2 and RUNX1 in a new supplementary figure: Fig S18).

R1 5: "Moreover, experiments combining single-cell ATAC-seq and RNA-seq should answer the question whether in the absence of functional TRIM28, FOXL2 indeed loses its capacity to repress the male pathway."

Response: Indeed, single-cell ATAC-seq experiments would provide very powerful information about the activation of the enhancer that can respond to the male pathway. However, it cannot show the differences of FOXL2 binding in the presence or absence of TRIM28.

We performed a ChIP-seq experiment for FOXL2 in cKO ovaries at week 8 when trans-differentiation is not achieved yet. We found a dramatic decrease in the number of FOXL2 peaks (from 50,000 peaks in WT to few thousands in cKO). However, it was difficult to interpret the decrease of peak number as the results of FOXL2 gene/protein extinction or as the loss of FOXL2 DNA binding due to the disappearance of TRIM28. A more suitable approach would be a single-cell CUT & RUN experiment to target FOXL2. However, such technology

is not easy to implement, and no commercial kits are available from 10x, and therefore, it would require a completely new study with substantial technology developments.

Reviewer #2 (Remarks to the Author):

The manuscript by Rossitto et al reports the latest example of an XX gonad masculinisation phenotype in the postnatal mouse ovary following gene deletion. Whilst this phenotype has previously been associated with transcription factor (TF) deletion – most notably FOXL2 – here the cause of sex reversal is loss of a protein involved in SUMOylation, TRIM28. Using some nice genetic approaches, the authors show that this SUMOylation activity is required to maintain ovarian somatic cell identity. They also use genome-wide approaches to show that SUMOylation appears to function at ovarian genes to stabilise their activity through impacts on FOXL2 and other key TFs. This study strengthens our understanding of the global role played by sex-determining genes in crafting an ovarian or testicular chromatin architecture and will be of interest to many in the field of sex determination, chromatin, post-translational modification and maintenance of cell fate more broadly. But the manuscript lacks some clarity in places and the comments below are aimed at trying to improve its clarity and impact.

Title: The title refers to the ‘male’ pathway. Given that women with complete androgen insensitivity syndrome have testes, and given how the classification of sex and gender is increasingly controversial, it is probably best not to identify sex with gonadal sex. The ‘testicular pathway’ or ‘Sertoli cell pathway’ would be preferable – and this is the case throughout the whole manuscript, where ‘male’ and ‘female’ are used to denote ‘testis’ and ‘ovary’ (R2_1).

Abstract: TRIM28 is said to ‘bind in close proximity’ – but how does an enzyme bind chromatin (R2_2)? It should be made clear early in the manuscript what sort of protein TRIM28 is (not DNA binding) and how recruitment to chromatin is presumably via proteins that can ultimately bind DNA (R2_3).

Also, why is it ‘remarkable’ that an E3 SUMO ligase functions to regulate SUMOylation (R2_4)?

Page 2, line 15: TRIM28 is a partner of testicular SOX9 – so isn’t unexpected that it functions to maintain the ovarian soma? Why is this ‘paradox’ not discussed (R2_5)? Do the SUMO peaks identified later in the manuscript coincide with SOX9 peaks following ChIPseq (R2_6)?

Page 2: lines 21-23 state that TRIM28 protein is not detected in cko gonads – is this at birth (R2_7)? How early is this loss seen (R2_8)? Can it be inferred that TRIM28 is not required for primary sex determination unless it is shown that TRIM28 is absent in fetal gonads during the sex determining stages (R2_9)? Perhaps loss is gradual and this contributes to a late onset phenotype (R2_10)?

P4, line 10: perhaps a little more explanation would be useful (especially for non-sex expert readers) why the medullary-cortical polarity in the appearance of sex reversal is explained by what is known about the wave-like appearance of granulosa cells during fetal development (R2_11).

P4: the use of ‘transdifferentiation’ and ‘reprogramming’ is first used here. Here (or in the conclusion section) it should be summarised briefly why the best explanation for the sex reversal is reprogramming/transdifferentiation, as opposed to, for example, loss of granulosa cells and the appearance of Sertoli-like cells from another (previously unrecognised) progenitor. The paper by Uhlenhaut et al (2009 Cell) that first reported this phenotype following Foxl2 ablation went to great lengths to exclude other explanations. Did the authors, for example, assess apoptosis in the cko gonads (R2_12)? Could loss of germ cells contribute to the sex reversal in this line (R2_13)? Why is transdifferentiation the only possible explanation (R2_14)?

P6: What is the explanation for why Rspo1 expression appears to be unaffected (R2_15)? RSPO1 is a major pro-WNT factor, so why is WNT disrupted in the cko (R2_16)?

P7: the term ‘pseudo-Sertoli cell’ is new and should be explained in further detail. Is it the same as ‘Sertoli-like’ (R2_17)?

P8: the distribution of cko cells shown in Fig 2a shows a few ‘intermediate’ cells, but the bulk of cko cells appear to be (spatially, so transcriptionally) closely associated with control ovarian somatic cells i.e. they form part of a larger ovarian cluster. So, the distribution still appears bi-modal whatever the genotype, apart from a small number of cells between ovarian and testicular control clusters. Is this expected for this sex reversal phenotype? Are there no Sertoli cells in the cko gonad? What are the cko granulosa cells transdifferentiating into? Some further clarification is important here. Just how sex reversed are these sex-reversed cko gonads (R2_18)?

P13: Do the PHD/PHD homozygotes die early, like the straight KO? This should be made clear (R2_19).

P15: Were any of the SUMO1/2 peaks occupied by RUNX1 (R2_20)? Given the Nicol et al (Nat Commun 2019) report of overlap of FOXL2 and RUNX1 chromatin occupancy, might TRIM28 SUMOylate both FOXL2 and RUNX1 (R2_21)? Was Cldn11 a site of hyper-SUMOylation (R2_22)?

P21: the manuscript ends with a comment about human fertility. But perhaps more interesting is whether there is any genetic evidence of a role for TRIM28 in human sex determination (like RSPO1)? There are a number of exome sequencing resources that could be examined. Or a role in female POI? Could the authors speculate (R2_23)?

Finally, it would be good for the authors to briefly clarify in the Discussion whether the activation of the testicular pathway is simply a consequence of the down-regulation of the ovarian pathway i.e. due to their mutual antagonism, or whether TRIM28 functions itself to directly oppose the testicular pathway, or both the above (R2_24).

Responses to reviewer #2

We thank reviewer 2 for the constructive remarks about the manuscript, and here are our answers to the comments.

R2 1: “Title: The title refers to the ‘male’ pathway. Given that women with complete androgen insensitivity syndrome have testes and given how the classification of sex and gender is increasingly controversial, it is probably best not to identify sex with gonadal sex. The ‘testicular pathway’ or ‘Sertoli cell pathway’ would be preferable – and this is the case throughout the whole manuscript, where ‘male’ and ‘female’ are used to denote ‘testis’ and ‘ovary’”.

Response: We agree with these comments and to exclude any potential controversial interpretation of our manuscript we corrected the title and throughout the manuscript we replaced male/female by testicular/ovarian. We only kept the expression “female-to-male sex reversal” a couple of times because it is currently used in the field.

R2 2: “Abstract: TRIM28 is said to ‘bind in close proximity’ – but how does an enzyme bind chromatin?”

Response: We think that the term “binds chromatin” is unambiguous because it describes proteins that interact with other proteins that constitute chromatin. Nevertheless, we cannot exclude that TRIM28 may binds directly to DNA. However, we changed the sentence “On chromatin, TRIM28 binds in the close proximity of FOXL2” that was unclear to “TRIM28 is recruited on chromatin in the proximity of FOXL2”. (Rev#2-1)

This point was added in the manuscript (Rev#2-2): We explain briefly that TRIM28 has been recently shown to interact with chromatin in ES cells using SPACemap (Rafiee et al, 2021. PMID: 34871434), a recently developed proteomic technique which allows identification of chromatin-binding proteins as well as their chromatin-contact regions. TRIM28 was shown to bind to chromatin through three unambiguous peptides: MAILQIMK (amino acids 298-305), FASWALESDNNTALLLSK (amino acids 349-366) and EEETEAAIGAPPAPEGPETKPVLMPLTEGPGAEGPR (amino acids 555 to 591). The first two peptides are in the RBCC domain and the third in an unfolded domain.

R2 3: “It should be made clear early in the manuscript what sort of protein TRIM28 is (not DNA binding) and how recruitment to chromatin is presumably via proteins that can ultimately bind DNA”.

Response: We added the following sentence at the beginning of the manuscript: “TRIM28 is a versatile nuclear scaffold protein that coordinates the assembly of protein complexes containing different chromatin remodeling factors. It can be recruited on chromatin upon interaction with DNA-binding proteins, such as KRAB-ZNF family members, or with transcription factors.” (Rev#2-3)

R2 4: “Also, why is it ‘remarkable’ that an E3 SUMO ligase functions to regulate SUMOylation?”

Response: Since its discovery at the end of the 1990s, TRIM28 function has been analyzed most of the time by in vitro and biochemical approaches and multiple functions have been assigned to TRIM28: transcriptional repression, HP1 recruitment on chromatin, and also E3 SUMO ligase activity. However, it has never been clearly demonstrated that the function of TRIM28 can be equated to its SUMO-ligase activity. For the first time we show using a genetic approach that the complete deletion of TRIM28 is recapitulated by the lack of SUMO-ligase

activity. For this reason, we used the term “remarkably”. However, we eliminated it from the abstract.

R2_5: “Page 2, line 15: TRIM28 is a partner of testicular SOX9 – so isn’t unexpected that it functions to maintain the ovarian soma? Why is this ‘paradox’ not discussed?”

Response: We modified the manuscript at this point (lines 21-23): “Despite its interaction with SOX9, cKO of *Trim28* in Sertoli cells results in adult males with hypoplastic testes with defect in spermatogenesis, but no sex reversal⁹. This suggests that TRIM28 is required in Sertoli cell for their role in the control of spermatogenesis but not for their maintenance”. (Rev#2-4) We also added a sentence at the end of the Discussion section: “In contrast with its role in granulosa cells, TRIM28 is not required for maintenance of adult Sertoli cells where it is involved in their crosstalk with germ cells⁹ and also appears as an important actor of SUMOylation⁶⁹.” (Rev#2-5)

R2_6: “Do the SUMO peaks identified later in the manuscript coincide with SOX9 peaks following ChIPseq?”

Response: This is a very interesting question that we omitted to discuss in the original manuscript. As our paper was submitted, David Zarkower’s group published an article in NAR (Lindeman et al 2021. PMID:34096593) where they analyzed the relationship between DMRT1 and SOX9 in a mouse model of ovarian reprogramming with ectopic expression of *Dmrt1*. Using the DMRT1 and SOX9 datasets provided in this paper, we observed overlapping between hyper-SUMOylated peaks and DMRT1 peaks (~16-18%). While only a smaller proportion overlapped with SOX9 peaks (~5%). Although DMRT1 SUMOylation has never been described, here we found that it is recruited to chromatin where de novo SUMOylation is detected. This may suggest a potential role as SUMOylated co-factor.

On the basis of the data by Lindeman and colleagues, compared with SOX9, DMRT1 should be the major contributor to transdifferentiation, which is not in contradiction with our data showing that compared with DMRT1, a minority of hyper-SUMOylated peaks contains SOX9. This analysis has been added in the manuscript as a new supplementary figure: Fig S20 (Rev#2-6)

R2_7: “Page 2: lines 21-23 state that TRIM28 protein is not detected in cKO gonads – is this at birth?”

Response: We are sorry for not having been clear in the manuscript. Effectively, it is soon after birth: at 3dpp, TRIM28 has disappeared in FOXL2-positive cells. (Rev#2-7)

R2_8: “How early is this loss seen?”

Response: To answer this question, we made additional experiments and modified Fig S1. We analyzed the expression of TRIM28 at E13.5, a stage where sex determination has already occurred, and at E18.5 prior to birth. At E13.5, in mutants, some TRIM28 protein is still visible in FOXL2+ cells as nuclear foci that we interpreted as being heterochromatin. At E18.5, TRIM28 has almost completely disappeared from pre-granulosa cells. All these results are now reported in Fig S1. (Rev#2-8)

R2_9: “Can it be inferred that TRIM28 is not required for primary sex determination unless it is shown that TRIM28 is absent in fetal gonads during the sex determining stages?”

Response: TRIM28 role in primary sex determination is an important question that we would like to answer in a future study. TRIM28 interacts indirectly through KRAB-0 with SRY (PMID: 19850934), suggesting that it may be involved in the testicular determining pathway. However, the fact that TRIM28 is still present at E13.5 did not allow us to conclude on its role

in primary sex determination. We discuss this point in the manuscript Rev#2-9 (Results and Discussion). The conditional knock-out should require a Cre-driver that is active very early during embryonic gonad formation, for instance *Wt1CreERT2* or *Osr1CreERT2* (Hu et al 2014, PMID: 23874227).

R2_10: “Perhaps loss is gradual, and this contributes to a late onset phenotype?”

Response: The observations that the TRIM28 signal has disappeared before birth in granulosa cells and that neither SOX8 nor SOX9 staining was detected at 3 dpp show that the transdifferentiation process occurs only after birth.

R2_11: “P4, line 10: perhaps a little more explanation would be useful (especially for non-sex expert readers) why the medullary-cortical polarity in the appearance of sex reversal is explained by what is known about the wave-like appearance of granulosa cells during fetal development.”

Response: This has been added in the Discussion Rev#2-10.

R2_12: “P4: the use of ‘transdifferentiation’ and ‘reprogramming’ is first used here. Here (or in the conclusion section) it should be summarized briefly why the best explanation for the sex reversal is reprogramming/transdifferentiation, as opposed to, for example, loss of granulosa cells and the appearance of Sertoli-like cells from another (previously unrecognised) progenitor. The paper by Uhlenhaut et al (2009 Cell) that first reported this phenotype following *Foxl2* ablation went to great lengths to exclude other explanations. Did the authors, for example, assess apoptosis in the cko gonads?”

Response: We investigated apoptosis in control and mutant ovaries at 20 dpp and 8wks using TUNEL assay. At both stages, we did not detect any significant increase of apoptotic cells in cKO ovaries compared with control where apoptosis is observed in atretic follicles and corpora lutea. Therefore, we can exclude widespread apoptosis of mutant granulosa cells followed by their replacement by de-novo differentiated Sertoli-like cells. This is in agreement with previous observations by Uhlenhaut et al in adult *FOXL2* knock-out and by Dupont et al, 2003, (PMID 12508230) in an estrogen receptor-double knock out. This has been added in a new supplementary Fig S4 and discussed in the manuscript Rev#2-11.

We think that the best explanation is the transdifferentiation because our sc-RNA-seq analysis shows clearly the different steps that granulosa cell undergo to become pseudo-Sertoli cells (see below Rev#2-12).

R2_13: “Could loss of germ cells contribute to the sex reversal in this line?”

Response: In our case, TRIM28 is not deleted in oocytes, and thus we can exclude that granulosa cells could be transdifferentiated upon expression of a hypothetical substance by oocytes. Therefore, as Uhlenhaut et al has clearly shown that loss of oocytes cannot explain the sex-reversal phenotype observed in *FOXL2* knock-out, we think that we can also exclude this hypothesis in the TRIM28 mutant.

R2_14: “Why is transdifferentiation the only possible explanation?”

Response: We think the single-cell data support transdifferentiation rather than loss of germ cells and independent appearance of Sertoli-like cells because we have cells that are transcriptionally ‘intermediate’ between the two mature cell states. Interestingly, this intermediate step probably exists in the *FOXL2* knock-out adults described by Uhlenhaut et al. Specifically, upon tamoxifen injection there is one day between *FOXL2* disappearance and the onset of *SOX9* expression when granulosa cells do not express *FOXL2* and *SOX9*. This double-

negative step might be the intermediate that we observed by scRNA-seq where neither FOXL2 nor SOX9 transcripts are expressed in the absence of TRIM28. This point is now discussed in the manuscript (Discussion) Rev#2-12.

R2_15: “P6: What is the explanation for why *Rspo1* expression appears to be unaffected?”

Response: Our ChIP-seq experiments provide an explanation to this observation. We did not find binding of FOXL2 and TRIM28 in the proximity of *RSPO1* in the ovary. Conversely, the FOXL2 ChIP-seq data from foetal ovary published by Nicol et al (PMID 30212841) show FOXL2 peaks 3' to *RSPO1* (chr4:125010336-125010635). Our data suggest that in the adult ovary, the *Rspo1* gene is not directly regulated by TRIM28 or FOXL2, a scenario that might be different in the foetal ovary.

R2_16: “*RSPO1* is a major pro-WNT factor, so why is WNT disrupted in the cKo?”

Response: Unlike *Rspo1*, the *Wnt4* gene displays several binding peaks for both TRIM28 and FOXL2 that explain its downregulation observed by RT-qPCR in the cKO. The profiles of *Rspo1* and *Wnt4* have been added in Fig S11, and these two points (R2_15 and R2_16) are now discussed in the manuscript Rev#2-13.

R2_17: “P7: the term ‘pseudo-Sertoli cell’ is new and should be explained in further detail. Is it the same as ‘Sertoli-like’?”

Response: We changed, in the manuscript, from “pseudo-Sertoli” to “Sertoli-like”.

R2_18: “P8: the distribution of cko cells shown in Fig 2a shows a few ‘intermediate’ cells, but the bulk of cko cells appear to be (spatially, so transcriptionally) closely associated with control ovarian somatic cells i.e., they form part of a larger ovarian cluster. So, the distribution still appears bi-modal whatever the genotype, apart from a small number of cells between ovarian and testicular control clusters. Is this expected for this sex reversal phenotype? Are there no Sertoli cells in the cko gonad? What are the cko granulosa cells transdifferentiating into? Some further clarification is important here. Just how sex reversed are these sex-reversed cKO gonads?”

Response: To perform a complete analysis of transdifferentiation, we would have needed to assess by scRNA-seq samples at week 8 and week 16 to cover the whole transdifferentiation process. Our priority has always been to investigate the molecular mechanisms underpinning transdifferentiation. In this perspective, it is preferable to analyze the initial alterations that induce transdifferentiation focusing on the early events, although there are only few or no completely transdifferentiated cells (i.e. Sertoli cells) in the cKO. For this reason, we performed the scRNA-seq analysis only at week 8. A scRNA-seq analysis at a later stage would have allowed us to better understand the consequences, but not the cause of transdifferentiation.

R2_19: “P13: Do the PHD/PHD homozygotes die early, like the straight KO? This should be made clear.”

Response: We never obtained PHD/PHD animals when crossing PHD/+ with PHD/+, therefore we strongly suspect that homozygotes might die during embryogenesis, like the straight KO. This has been clarified in the manuscript. (Rev#2-14)

R2_20: “P15: Were any of the SUMO1/2 peaks occupied by RUNX1?”

Response: It is difficult to answer this question because the ChIP-seq data by Nicol et al were obtained using foetal ovaries. The number of peaks obtained in foetal ovaries is much lower than in adult ovaries: for instance, we obtained 58,581 peaks for FOXL2 in adult ovaries and

Nicol et al 11,438 peaks for FOXL2 and 10,494 for RUNX1 in foetal ovaries. This difference can depend on multiple parameters (e.g. antibody used, amount of material, sequencing depth).

Nevertheless, by looking at the overlap of foetal FOXL2 and RUNX1, we observed that only a small proportion (~1%) of hypoSUMOylated peaks in post-natal mutants is occupied by FOXL2 and RUNX1 in foetal ovary, while 30 to 40% are occupied by FOXL2 in adult ovaries. (see the figure below)

This makes it difficult to conclude on the possibility that hypoSUMOylated peaks might represent SUMOylated RUNX1 on chromatin in wild type ovaries. Therefore, only a ChIP-seq experiment on RUNX1 in adult ovaries might answer this question. This is discussed in the manuscript. However, by taking advantage of the article by Zarkower's group, we compared their ESR2 ChIP-seq datasets (obtained using adult ovaries) with our data, and we found that hypo-SUMOylated peaks display an overlap with ESR2, but not as strong as the overlap with FOXL2 (Rev#2-15).

R2 21: "Given the Nicol et al (Nat Commun 2019) report of overlap of FOXL2 and RUNX1 chromatin occupancy, might TRIM28 SUMOylate both FOXL2 and RUNX1?"

Response: We provide a figure (Fig S18) to address this point. As we answered to reviewer 1, it is not a trivial experiment to perform from tissue samples and obtaining the complete SUMOylome for control and mutant ovaries would be a new study on its own. However, we wanted to determine whether TRIM28 might SUMOylate FOXL2 and RUNX1. To do so, we performed an in vitro/in cell SUMOylation assay where we transfected HEK293 cells with expression plasmids for V5-tagged FOXL2 and RUNX1, together with HA-SUMO2, in the presence or absence of expression plasmids for TRIM28 or TRIM28 C651F (the PHD mutant). We immunoprecipitated V5-tagged FOXL2 and RUNX1 proteins and visualized SUMO conjugation using an anti-HA antibody. This result is presented in Fig S18. Using this technique, we observed that TRIM28 (but not the PHD mutant) induces the specific SUMOylation of FOXL2 and RUNX1. Therefore, both proteins are potential targets of the E3-SUMO ligase activity of TRIM28 in vivo. This is now discussed in the manuscript (Rev#2-16).

R2 22: "Was *Cldn11* a site of hyper-SUMOylation?"

Response: This is a very interesting question that we did not address in the manuscript. *Cldn11* is one of the earliest markers of transdifferentiation toward the Sertoli cell fate, as we observed by single-cell RNA-seq. Looking at the ChIP-seq data, we found that the upstream region of the *Cldn11* gene displays multiple peaks for FOXL2 and TRIM28, suggesting a repressive effect of these two factors on *Cldn11* expression. Moreover, this region also presents a peak of hyperSUMOylation in the cKO and PHD mutants. This hyperSUMOylated peak is also bound

by TRIM28 and FOXL2 in control, suggesting that in their absence, this region might be open for binding by some Sertoli-specific transcription factors that are SUMOylated in both mutants. This is also an open chromatin region, as observed by ATAC-seq in embryonic gonads (Garcia-Moreno et al 2018, PMID 30594505). We added this point in the Results section of the manuscript and we enlarged the corresponding panel in Figure S12 (ChIP-seq TRIM28 and FOXL2) and added Fig S23 (ChIP-seq SUMO). (Rev#2-17)

R2 23: “P21: the manuscript ends with a comment about human fertility. But perhaps more interesting is whether there is any genetic evidence of a role for TRIM28 in human sex determination (like RSPO1)? There are a number of exome sequencing resources that could be examined. Or a role in female POI? Could the authors speculate?”

Response: This has been integrated in the Discussion section (Rev#2-18).

R2 24: “Finally, it would be good for the authors to briefly clarify in the Discussion whether the activation of the testicular pathway is simply a consequence of the downregulation of the ovarian pathway i.e. due to their mutual antagonism, or whether TRIM28 functions itself to directly oppose the testicular pathway, or both the above.”

Response: We discussed this point in the Discussion section. TRIM28 must interact with a transcription factor to exert its effect on gene regulation. Therefore, it is within the framework of an ovarian environment that it can exert its repression on the testicular program. Conversely, TRIM28 activity in the testis is different because its ablation in adult Sertoli cells does not induce transdifferentiation (Herzog et al 2011, PMID 21163256; and our unpublished observation). Its ablation only alters the function of Sertoli cells, visualized by germ cell defects, but it does not change cell fate. Therefore, it appears that TRIM28 role is very different in ovarian and testicular supporting cells. This has been added in the Discussion section (Rev#2-19).

Reviewer #3 (Remarks to the Author):

The manuscript by Rossitto et al identifies a new role of TRIM28 in the maintenance of the postnatal ovary identity. Conditional KO of Trim28 in mouse ovary leads to ovary-to-testis sex reversal and scRNA-seq demonstrates transdifferentiation of granulosa cells into an intermediary state resembling testis Sertoli cells. The Trim28 cKO phenotype was similar to Foxl2 postnatal KO phenotype. ChIP-seq for TRIM28 in adult ovary revealed overlap with FOXL2 ChIP-seq, suggesting that TRIM28 may act along with FOXL2 or maybe? through posttranslational regulation of FOXL2 to maintain ovarian identity. TRIM28 has indeed an intrinsic E3 SUMO ligase activity. The authors created a point mutation mouse model that abrogates this function, and the phenotype of these mice was similar to that of the Trim28 cKO, indicating that TRIM28 acts through this SUMO ligase function to maintain ovarian identity. Comparison of SUMO1-2/3 ChIP-seq between control ovary and both Trim28 mutant models revealed

significant changes in sumoylation in the mutant ovaries, including around granulosa/Sertoli specific genes.

Overall, this is a well-written manuscript, with judicious use of various transcriptomic / genomic techniques and mouse models to characterize in details the functions of TRIM28 in the ovary.

Here are my comments:

My main comment relates to the role TRIM28 and sumoylation in transcriptional regulation. TRIM28 and sumoylation are commonly associated with transcriptional repression. How do you reconcile this with your findings that genes downregulated are hyposumoylated whereas genes upregulated are hypersumoylated? This seems to go against the consensus of hyposumoylation is associated with transcriptomic activation. This needs to be further discussed (R3 1).

Since TRIM28 is mostly known as a transcriptional repressor, what is the first event that occurs in the transdifferentiation of cKO ovaries? Upregulation of male genes, or downregulation of female genes? Sox8 transcript levels are already high at 0.5m in Trim28 cKO ovaries (Fig. 1C), to the level of control testes. Could de-repression of some Sertoli genes happen before the downregulation of granulosa genes in cKO ovaries (R3_2)?

Have you looked at an earlier time-point than 20 days if SOX8 protein expression precedes the loss of FOXL2 protein rather than appearing after FOXL2 loss in supporting cells (R3_4)?

In Fig. 1A, the authors show that SOX8 appears before SOX9. There are however some discrepancies between immunos at 20d and 8w (Fig. S3 bottom). While at 20d there seem to be more SOX8+ cells than SOX9+ cells (Fig 1A), at 8w old, there are more SOX9 single positive cells in follicles than SOX8 single positive cells. How do you explain this difference (R3_5)?

Similarly, there are some discrepancies in FOXL2 labelling in 8-week-old ovaries in Fig. S3. In the double immunostaining FOXL2/SOX8, it is very clear that some follicular cells are still expressing FOXL2 in most follicles in the medulla, particularly at the periphery of the follicles. However, in the below double immunostaining FOXL2/SOX9, FOXL2 appears to be gone from most follicles in the medulla. Is there a lot of inter-animal variation in the phenotype? Which phenotype is more common (R3_6)?

It is actually quite interesting that the double labeling FOXL2/SOX8 shows that almost all the supporting cells in the one layer at the periphery of the follicle / pseudo-tubular structure maintain FOXL2 expression (Fig S3). Could these cells be more resistant to transdifferentiation (R3_7)?

Why do you think some cKO supporting cells transdifferentiate while others seem to maintain a pretty normal granulosa transcriptome based on scRNA-seq? Is it the fetal origin of the 2 waves of granulosa cells, medulla/cortex that prime them into their different capacity to respond to masculinization signals (R3_8)?

FOXL2 functions, for instance transcriptional repression, and stability rely on its sumoylation state. TRIM28 cKO phenotype is similar to FOXL2 postnatal KO. Do you think TRIM28 acts solely on FOXL2 to maintain ovarian identity or does it play a more global genomic role / targeting other TFs involved in supporting cell identity as well (R3_9)? Have you looked at FOXL2 protein sumoylation state in ovaries of the two TRIM28 mutant models (R3_10)?

The authors mention that heterozygous Nr5a1:Cre;Trim28 flox/+ have no phenotype. Since these mice are used for the cross with the Trim28 C651F mutants to characterize the effects of the sole expression of this mutated Trim28 protein in the ovary, it is relevant to provide the data proving normal ovarian phenotype, such as PAS staining in Fig S13. Have the authors looked at gene expression to confirm that haploinsufficiency of Trim28 in Nr5a1 cells does not affect granulosa/Sertoli genes (even though these mice are fertile), and potentially contributes to the PHD/cKO mouse phenotype (R3_11)?

Fig 5C: The tracks for TRIM28 and FOXL2 ChIP-seq need to be directly added in order to easily compare with all the SUMO tracks. Going back and forth with Fig.3C, which has different track sizes, is not ideal (R3_12).

Throughout the manuscript, the authors state that granulosa cells are reprogrammed into Sertoli cells. However, the single cell RNA-seq analysis clearly shows that these transdifferentiated cells never reach a full Sertoli cell state as they do not cluster with the normal testis Sertoli cell and rather stay in an intermediary state between granulosa and Sertoli identity (R3_13). It would therefore be more accurate to call these cells "Sertoli-like" rather than Sertoli (R3_14).

Responses to reviewer #3

We thank reviewer 3 for the constructive remarks about the manuscript, and here are our answers to the comments.

R3 1: “My main comment relates to the role TRIM28 and sumoylation in transcriptional regulation. TRIM28 and sumoylation are commonly associated with transcriptional repression. How do you reconcile this with your findings that genes downregulated are hyposumoylated whereas genes upregulated are hypersumoylated? This seems to go against the consensus of hyposumoylation is associated with transcriptomic activation. This needs to be further discussed.”

Response: Already for long time, both TRIM28 and SUMOylation have been associated with transcriptional repression. However, this consensus has been challenged:

-several papers showed that TRIM28 is a co-activator of C/EBPbeta and glucocorticoid receptor (Chang et al 1998, PMID 9742105), of the orphan nuclear receptor NGFI-B/Nur77 (Rambeaud et al 2009, PMID 19321449). Genome-wide, TRIM28 activates transcription of some of its target genes (Iyengar et al 2011, PMID 21343339) and also controls transcriptional pausing (Bunch et al 2014 PMID 25173174) and elongation through the recruitment of the 7SK snRNP complex (McNamara et al 2016, PMID 26725010). Moreover, TRIM28 is a transcriptional activator of the mutant TERT promoter in bladder cancer (Agarwal et al 2021, PMID 34518220).

This point is now added in the manuscript (Rev#3-0).

Globally TRIM28 can be seen as a nuclear scaffold involved in every aspect of gene regulation: activation, pausing, and repression.

Concerning, SUMOylation, the answer could be more or less the same. This post-translational modification has been thought for long time as a repressive event of transcription factors (TF) activity toward transcription. However, this rule is now challenged:

-SUMOylation of TFs enhances their binding site selection by increasing their specificity toward their DNA binding site, and thus it might increase their transcriptional activity (Sri Theivakadacham et al 2019, PMID: 30763307; Rosonina 2019, PMID: 31093693).

- SUMOylation potentiates the transcriptional activity of a growing number of TFs (PAX6, GATA-1, GATA-4, SMAD4, OCT4,...for a review see Boulanger et al 2021, PMID: 33562565). For example, in vivo mutation of SUMOylated lysine residues on the androgen receptor induces dysregulation of its target genes in epididymis with up- or down-regulation (Zhang et al, 2019, PMID 30770815). This shows that SUMOylation effects on TF activity largely depend of the gene context.

-A recent genome-wide study showed that SUMOylation controls adipogenesis through positive regulation of PPAR γ and RXR on their target genes (Zhao et al 2022, PMID: 35100417).

Therefore, our observations are part of an evolution on the understanding of SUMOylation in the regulation of transcription. Discussed in Rev#3-2.

In the revision process of this manuscript, we showed that TRIM28 induces SUMOylation of FOXL2 when expressed in HEK293 cells (as these results are preliminary and require further investigation, we provide them as supplementary data: Fig S18). Previous studies showed that SUMOylation is mainly required to stabilize this TF and therefore, it may enhance it

transcriptional activation capacity (Georges et al 2011. PMID 22022399, Kim et al 2014 PMID: 24390485). However, in a different gene context, it can also enhance its repressing capacity (Marongiu et al 2010, PMID 20209145). Altogether, these results suggest that SUMOylation effect on FOXL2 activity is dependent on the gene context.

R3 2: “Since TRIM28 is mostly known as a transcriptional repressor, what is the first event that occurs in the transdifferentiation of cKO ovaries?”

Upregulation of male genes, or downregulation of female’s genes? Sox8 transcript levels are already high at 0.5m in Trim28 cKO ovaries (Fig.1C), to the level of control testes. Could de-repression of some Sertoli genes happen before the downregulation of granulosa genes in cKO ovaries?”

Response: As we said before, TRIM28 can work either as a transcriptional repressor or activator, depending on the gene context.

Using sc-RNA-seq we found that some genes, such as *Cldn11*, are progressively activated (or de-repressed) and are expressed in cells that still express the granulosa transcriptome and clearly before the upregulation of testicular master regulators, such as *Dmrt1* or *Sox9*. This can be explained by the loss of repressing activity of master transcription factors, such as FOXL2. We discussed this point in manuscript: Rev#3-3. At this level it is difficult to clearly determine what happens without additional genetic experiments (e.g. double knock-outs). However, we did not observe any measurable expression of *Sox8* in granulosa cells.

R3 4: “Have you looked at an earlier time-point than 20 days if SOX8 protein expression precedes the loss of FOXL2 protein rather than appearing after FOXL2 loss in supporting cells?”

Response: In our immunofluorescent experiments, we never saw any cell expressing both FOXL2 and SOX8, as reported in the manuscript Rev#3-4.

R3 5: “In Fig. 1A, the authors show that SOX8 appears before SOX9. There are however some discrepancies between immunos at 20d and 8w (Fig. S3 bottom). While at 20d there seem to be more SOX8+ cells than SOX9+ cells (Fig 1A), at 8w old, there are more SOX9 single positive cells in follicles than SOX8 single positive cells. How do you explain this difference?”

Response: We thank reviewer 3 to highlight this point of weakness in our manuscript because the data we provided in the first version of the manuscript in Fig S3 were not the most representative pattern at week 8. We re-analyzed our previous data and made new experiments on mutant ovaries from five different animals at this time-point. We obtained clear answers to this question.

At week 8, the number of SOX8+ cells (in red) is higher than that of SOX8+/SOX9+ or SOX9+ (in green) in the cortical regions (Fig S3 panel i) where follicular structures are still present. Conversely, in medullar regions where pseudo-tubules are more represented than follicular structures, pseudo-tubules contain SOX8+/SOX9+ cells and even SOX9+ cells (Fig S3 panel h).

This is in good agreement with our qPCR and scRNA-seq data and our immunofluorescence results from 20-day-old mutants (fig 1A), suggesting that SOX8 precedes SOX9 in the cascade leading to Sertoli-like cells. We provide a new version of Fig S3 with new SOX8/SOX9 double staining that is representative of what we saw in five different mutant animals. Rev#3-5

R3 6: “Similarly, there are some discrepancies in FOXL2 labelling in 8-week-old ovaries in Fig. S3. In the double immunostaining FOXL2/SOX8, it is very clear that some follicular cells are still expressing FOXL2 in most follicles in the medulla, particularly at the periphery of the follicles. However, in the below double immunostaining FOXL2/SOX9, FOXL2 appears to be gone from most follicles in the medulla. Is there a lot of inter-animal variation in the phenotype? Which phenotype is more common?”

Response: As we corrected the problem of SOX8/SOX9 double staining in Fig S3, we think that our results on FOXL2 double staining with SOX8 or SOX9 are now fitting well with the fact that SOX8 would precede SOX9 during transdifferentiation (supported by SOX8/SOX9 double staining, qPCR and sc-RNA-seq). Cells expressing only SOX8 might be less engaged in transdifferentiation, and this would explain why some cells are still expressing FOXL2 in the same follicle. Cells expressing SOX9 might be more engaged in transdifferentiation, explaining why most SOX9-positive medullar pseudo-tubules do not express FOXL2 any longer.

R3 7: “It is quite interesting that the double labeling FOXL2/SOX8 shows that almost all the supporting cells in the one layer at the periphery of the follicle / pseudo-tubular structure maintain FOXL2 expression (Fig S3). Could these cells be more resistant to transdifferentiation?”

Response: As we said before and regarding our other data, these SOX8/FOXL2-positive structures would reflect less transdifferentiated structure.

R3 8: “Why do you think some cKO supporting cells transdifferentiate while others seem to maintain a pretty normal granulosa transcriptome based on scRNA-seq? Is it the fetal origin of the 2 waves of granulosa cells, medulla/cortex that prime them into their different capacity to respond to masculinization signals?”

Response: This is a very relevant question, namely why some cells start to transdifferentiate very early while others remain unaltered. It is commonly assumed that most transcriptional processes are asynchronous, so this is not unique to our case. This is the basis of lineage-trajectory analysis with scRNA-seq- without the asynchronous programs, we would not be able to infer such trajectories from a sample taken at a single time point.

We think this heterogeneity is related to the cell microenvironment. Different local factors could affect the transdifferentiation process. Moreover, medullar follicles are more sensitive to the masculinization because we observed a strong difference between medulla and cortex at week 8. As these follicles are mostly derived from embryonic bi-potential precursors, this would suggest that they are more “unstable”. In follicles, our immunofluorescent experiments show that some isolated cells, randomly distributed among follicles, are switching to masculinization in cortical follicles, after the process appears to have spread to the rest of the follicle. Therefore, determining the origin of these “initiators” to see whether they originate from the first or the second wave of pre-granulosa would require lineage tracing experiments that might be the topic of a future study. We discussed this point in the manuscript (Rev#3-6).

R3 9: “FOXL2 functions, for instance transcriptional repression, and stability rely on its sumoylation state. TRIM28 cKO phenotype is similar to FOXL2 postnatal KO. Do you think TRIM28 acts solely on FOXL2 to maintain ovarian identity, or does it play a more global genomic role / targeting other TFs involved in supporting cell identity as well?”

Response: From our experiments we saw that some de-repressed genes are not bound by FOXL2: 1,633 TRIM28 peaks out of 51,764 do not overlap with FOXL2 peaks and correspond to 1,085 genes that are upregulated in cKO. From those genes, 671 are not bound by FOXL2. This suggest that other TFs might be involved together with TRIM28 in their repression. Among the TFs crucial for granulosa cell identity, there are the estrogen receptors ESR1 and 2

(Couse et al 1999, PMID 10600740) and recently RUNX1 (Nicol et al 2019, PMID 31712577). As ESR2 is the major estrogen receptor expressed in granulosa cells, both ESR2 and RUNX1 might be functionally interacting with TRIM28, like FOXL2. This point is discussed in the manuscript.

Moreover, as an ESR2 ChIP-seq experiment in adult ovary has been recently published (Lindeman et al 2021, PMID 34096593), we took advantage of these data to evaluate the overlapping of ESR2 with the hypo-SUMOylated peaks. We found an overlap that is not as strong as the one observed with FOXL2, but remains significant, with hypo-SUMOylated peaks obtained in both mutants. This has been added in Sup Fig S17. Rev#3-7 and Rev#3-8 Conversely, it was difficult to make this analysis for RUNX1 because only ChIP-seq data on foetal gonads are available, therefore it would be difficult to conclude.

R3 10: “Have you looked at FOXL2 protein sumoylation state in ovaries of the two TRIM28 mutant models?”

Response: This is a very relevant question because FOXL2 appears to be an evident target for SUMOylation by TRIM28, especially as some hypoSUMOylated peaks in both mutants are located on regions where FOXL2 normally binds to in the control. However, this not a trivial experiment to realize in vivo because SUMOylation is an extremely labile modification, and such approach is really tissue-consuming. We plan to do it in a future study, particularly investigating the SUMOylome in control and mutant ovaries.

Therefore, to answer this question, we determined whether TRIM28 could SUMOylate FOXL2 in vitro, by transfecting HEK293 cells. As shown in the new supplementary figure (Fig S18), we transfected a FOXL2 expression vector together with TRIM28 WT or TRIM28 C651F (the PHD mutant) and HA-SUMO2. After immunoprecipitation of FOXL2 using V5 tag in denaturing conditions (to avoid any co-precipitation of FOXL2 with SUMOylated proteins), we observed that FOXL2 is natively SUMOylated in HEK293 cells by endogenous SUMO-ligase (band at 58 kDa detected by an anti-HA antibody against HA-SUMO2 at the same MW as FOXL2 detected by the anti-V5 antibody).

This is in agreement with the calculated MW of FOXL2 encoded by the expression plasmid that should be 43.6 kDa while the observed FOXL2 band is 58 kDa. In the presence of TRIM28, we detected a band of higher MW (around 90kDa) that might correspond to the conjugation of a SUMO2 protein on a different lysine or a polySUMO chain on the same residue. This SUMOylated upper band of 90 kDa has been already observed in KGN cells, a human granulosa cell line (Kim et al 2021, PMID 24390485) that expresses FOXL2. Conversely, in cells co-transfected with FOXL2 and TRIM28 C651F (that lacks the SUMO-E3 ligase) we did not detect this 90 kDa SUMOylated form of FOXL2. Altogether, our results support the hypothesis that FOXL2 is SUMOylated by TRIM28. We also found that TRIM28 SUMOylates RUNX1 (a question by reviewer #2).

These results are presented in the new supplementary Figure S18 (Rev#3-9)

R3 11: “The authors mention that heterozygous *Nr5a1:Cre;Trim28 flox/+* have no phenotype. Since these mice are used for the cross with the *Trim28 C651F* mutants to characterize the effects of the sole expression of this mutated *Trim28* protein in the ovary, it is relevant to provide the data proving normal ovarian phenotype, such as PAS staining in Fig S13. Have the authors looked at gene expression to confirm that haploinsufficiency of *Trim28* in *Nr5a1* cells does not affect granulosa/Sertoli genes (even though these mice are fertile), and potentially contributes to the PHD/cKO mouse phenotype?”

Response: This is a good question because a recent study showed that mice lacking one *Trim28* allele display infertility (Tan et al, 2020. PMID 32302554).

To answer this question, we analyzed by immunostaining at three different stage (20 dpp, 4 and 8 weeks) the expression of FOXL2 and TRIM28 in WT (*Trim28^{+/+}; Nr5a1-Cre*), heterozygous (*Trim28^{lox/+}; Nr5a1-Cre*), and homozygous (*Trim28^{lox/lox}; Nr5a1-Cre*) ovaries for the mutant allele in the presence of the Cre transgene. As shown in Fig S6, we did not observe any difference between WT and heterozygous ovaries in FOXL2 expression and follicular organization. Conversely, in homozygous ovaries, FOXL2 expression was clearly decreased already at 4 weeks.

We wanted to test the expression of Sertoli markers (*Sox9, Sox8 and Dmrt1*) in these different genotypes. RT-qPCR analysis of 3-month-old ovarian samples did not show any significant increase of those genes in heterozygous animals compared with wild type. Conversely, in homozygous ovaries they were strongly upregulated (Fig S6b). From these observations we conclude that *Trim28* haploinsufficiency does not affect the granulosa cell fate. Reported in Fig S6 and discussed in the manuscript: **Rev#3-10**.

Moreover, we did not observe any defect in the capacity of heterozygous mutants to reproduce (number of pregnancies and litter size).

R3 12: “Fig 5C: The tracks for TRIM28 and FOXL2 ChIP-seq need to be directly added in order to easily compare with all the SUMO tracks. Going back and forth with Fig.3C, which has different track sizes, is not ideal.”

Response: We agree that doing a continual comparison of Fig 3C and Fig 5C to localize the position of FOXL2 and TRIM28 peaks relatively to the SUMO peaks is not ideal and confusing for the reader. However, with the size constraint for publication, providing TRIM28 and FOXL2 tracks together with SUMO tracks would have required to reduce the number of examples in this figure (and the related supplementary figures). Therefore, to facilitate the reading, we added a clear mark for each gene reported showing on SUMO tracks the localization of TRIM28 peaks (blue triangle), FOXL2 (red triangle) or both (double blue and red triangles).

R3 13: “Throughout the manuscript, the authors state that granulosa cells are reprogrammed into Sertoli cells. However, the single cell RNA-seq analysis clearly shows that these transdifferentiated cells never reach a full Sertoli cell state as they do not cluster with the normal testis Sertoli cell and rather stay in an intermediary state between granulosa and Sertoli identity.”

Response: We agree with the reviewer in that to visualize cells where the transdifferentiation process has completed we would need to perform scRNA-seq at week 16 (or later), covering the entire transdifferentiation process. However, our priority has always been to investigate the molecular mechanism at the source of transdifferentiation. In this perspective, it is preferable to analyze the initial alterations that induce transdifferentiation by focusing on the early events, although there are few or no completely transdifferentiated cells (i.e. Sertoli cells). For this reason, we performed a scRNA-seq analysis only at week 8. In fact, a scRNA-seq analysis at a later stage would have allowed us to better understand the consequences, but not the cause of transdifferentiation.

R3 14: “It would therefore be more accurate to call these cells “Sertoli-like” rather than Sertoli.”

Response: We agree with reviewer 3 and this has been corrected in the manuscript.

Reviewer #4 (Remarks to the Author):

This is a well-written manuscript describing a series of experiments establishing the role of TRIM28 in maintaining the female characteristics of the mouse ovary. The authors created a TRIM28 knock-out mouse (cKO) and observed progressive female-to-male transition in tissue structure, cell states, and known male/female markers. At multiple time points after birth the cKO ovaries displayed gradual conversion of the female-specific granulosa cells to a transcriptional state similar to the male-specific Sertoli cells, as shown by the molecular markers for the male and female lineages. Single-cell RNAseq data showed that the transdifferentiation did not produce bona fide Sertoli cells, rather an intermediate cell type between granulosa and Sertoli.

Starting from Figure 3, the study delved into detailed molecular mechanisms, focusing on the interaction between TRIM28 and FOXL2. ChIP-seq peaks for TRIM28 tend to coincide with those for FOXL2 (55%-62% overlap), suggesting that TRIM28 and FOXL2 jointly maintain the female fate, and TRIM29-cKO resulted in the upregulation of a host of male-specific genes. These data are consistent with prior studies of female-male transdifferentiation using other models, such as targeting FOXL2 or ESR1-2 in adult females.

The last part of the study tested the idea that the role of TRIM28 is mediated by SUMOylation. The authors generated a line carrying a point mutation in TRIM28 (called PHD) that removes its SUMO-E3 ligase activity and found that PHD/cKO and cKO ovaries are similarly affected. This result indicates that TRIM28 maintains the female state by its E3-SUMO ligase function. Protein localization/quantification by microscopy and scRNAseq experiments added more details to the comparison of PHD/cKO and cKO.

I have a few minor comments.

1. In Fig. S2 the oocytes positive for TRIM (red) in cKO ovaries are significantly larger than the wild type. It is not clear how to explain that (R4 1).
2. All the findings are from antral follicles. It's possible that the expression (or lack of it) of TRIM becomes important at the later stages of folliculogenesis (R4 2). It would be interesting to see if this mutation affects follicular reserve. At what stage of folliculogenesis does stabilization become important (R4 3)?
3. Fig. S4 - deposition "of basal laminae (green arrow)" is better determined with IHC. Is the composition in the mutant ovary different from the control (R4 4)?
4. It would be useful to clearly state how the Theca cells are affected by cKO. In Fig. S6 the insert is too small to see. It seemed that Clusters 23 and 27 are Theca cells (R4 5). The number of cells by cluster and by condition can be reported (R4 6).
5. In Fig. 5A and 5B, are the genes shown in the same order across the three panels: Cont, cKO, and PHD? That is, is each row for the same gene (R4 7)?
6. The data in GEO is not accessible. If it is temporarily blocked, there should be a reviewer token for access (R4 8).

Responses to reviewer #4

We thank reviewer 3 for the constructive remarks about the manuscript, and here are our answers to the comments.

R4 1: “1. In Fig. S2 the oocytes positive for TRIM (red) in cKO ovaries are significantly larger than the wild type. It is not clear how to explain that.”

Response: We thank reviewer#4 for this observation. We can hypothesize that granulosa cells are producing some signal that might regulate the oocyte growth and therefore that this might be deregulated in the absence of TRIM28 in granulosa cells. We added this observation in the manuscript: Rev#4-1.

R4 2: “2. All the findings are from antral follicles. It's possible that the expression (or lack of it) of TRIM becomes important at the later stages of folliculogenesis.”

Response: We do not think that transdifferentiation occurs only in antral follicles, for examples in Fig S2 cells expressing SOX9 are present in pre-antral follicles. Similarly, some cells expressing both SOX8 and SOX9 are present in smaller follicles that may be secondary follicles. Up to now, we have not observed a particular stage where the transdifferentiation should start preferentially.

R4 3: “It would be interesting to see if this mutation affects follicular reserve. At what stage of folliculogenesis does stabilization become important?”

Response: This would be an important point to investigate using scRNA-seq at different stages of transdifferentiation during the sexual maturation of mutant ovaries. Such experiments would require a high number of cells, and we believe this would constitute a whole new study. However, we can partially answer this question because we did not detect any organized follicle that expresses FOXL2 in 4-month mutant ovaries. This shows that growing follicles are affected by transdifferentiation and also the follicular reserve (where granulosa cells also express FOXL2).

R4 4: “3. Fig. S4 - deposition "of basal laminae (green arrow)" is better determined with IHC. Is the composition in the mutant ovary different from the control?”

Response: From our RNA-seq data, we saw that many gene encoding collagen proteins are upregulated in mutant ovaries: *Col6a5*, *Col4a3*, *Col28a1*, *Coll3a1*, *Col8a2*, *Col6a6*, *Col22a1*, *Col9a3*. Among them, *Col4a3*, *Col9a3*, *Coll3a1*, *Col28a1* encode components of collagen 4, 9, 13 and 28 that participate in extracellular matrix and basement membrane. Similarly, *Lamc2*, which encodes the laminin gamma2 chain that participates in basal lamina, also is upregulated in mutants. All these results suggest that the basal lamina is modified in mutant ovaries. (Rev#4-2)

R4 5: “4. It would be useful to clearly state how the Theca cells are affected by cKO. In Fig. S6 the insert is too small to see. It seemed that Clusters 23 and 27 are Theca cells.”

Response: It is a good question, but we did not see evidence of transdifferentiation in the steroidogenic lineage (i.e. no clear gradient of Theca to Leydig cells in the cKO). We do not know whether this is expected, but we guess that supporting cells change first, so perhaps 8 weeks (the time point of single cell sequencing) is too early to observe changes in the steroidogenic lineage.

R4 6: “The number of cells by cluster and by condition can be reported.”

Response: This has been added in the excel file “Data S2”.

R4 7: “5. In Fig. 5A and 5B, are the genes shown in the same order across the three panels: Cont, cKO, and PHD? That is, is each row for the same gene?”

Response: We agree that heatmaps are not very easy to understand, and we modified Figures 5A and 5B. We now use a boxplot showing in Fig 5a the quantification of SUMO1 or SUMO2 sequencing reads at deregulated regions in control, cKO and PHD. The upper part (blue) shows the quantification of hypo-SUMOylated regions, and the lower part shows the hyper-SUMOylated regions. Figure 5B shows the number of SUMO1/2 sequencing reads that are located at the TRIM28 peaks: in blue, the reads that decreased in mutant (Hypo-SUMO) and in red the reads that increased in mutants (hyper-SUMO). The goal of Fig 5B is to show that in mutants the decrease is more frequent than the increase in SUMOylation on TRIM28 peaks.

R4 8: “6. The data in GEO is not accessible. If it is temporarily blocked, there should be a reviewer token for access.”

Response: We are sorry that reviewer 4 missed this information. We deposited the relevant high-throughput data generated in our manuscript at the GEO public database under two accession numbers: GSE166385 (RNA-seq and ChIP-seq) token: uxqzaocinjcjdin and GSE166444 (single-cell sequencing) token: wvgvewyyjbivned, and at ProteomeXchange public database under accession number PXD024439 (mass spectrometry proteomics), username: reviewer_pxd024439@ebi.ac.uk, password: 3CvA03c6.

REVIEWERS' COMMENTS

Reviewer #1 (Remarks to the Author):

The authors have alleviated my concerns in a satisfactory manner.

Reviewer #3 (Remarks to the Author):

In their revised manuscript and rebuttal, the authors have addressed my concerns by providing additional experiments, modifying their figures and the text. All the efforts provided are appreciated and improve the manuscript.

There is only one disagreement that should be revised in the text of the final manuscript version:

R3_4: "Have you looked at an earlier time-point than 20 days if SOX8 protein expression precedes the loss of FOXL2 protein rather than appearing after FOXL2 loss in supporting cells?" Response: In our immunofluorescent experiments, we never saw any cell expressing both FOXL2 and SOX8, as reported in the manuscript Rev#3-4.

I disagree with this response. Based on Fig.1a, it is noticeable that at 20 dpp, many SOX8+ supporting cells are FOXL2 / SOX8 double positive, with a gradient of overlap transitioning from yellow/light green (high FOXL2, low SOX8) to dark orange (low FOXL2, high SOX8). See attached image for Fig. 1a with added arrows showing examples of yellow or orange double positive cells within the follicles.

This suggests that some supporting cells start expressing SOX8 before FOXL2 is fully turned off. Then by 8 w (Fig S3a), they are indeed mostly mutually exclusive as the authors mentioned in their rebuttal response. This observation provides input on the transdifferentiation event, which clearly is ongoing at 20 dpp, answering the question R2_12 from reviewer 2 that the Sertoli-like cells indeed originate from the differentiated granulosa cells within follicles, rather than from another source. This is also in agreement with other publications describing supporting cell transdifferentiation events in mouse gonads, for which a brief/transient co-expression of granulosa/Sertoli markers was observed. The presence of these double positive cells should be mentioned in the manuscript.

Other minor comment:

Fig. 5d: track information is too small to read (y axis, chromosomal position, gene orientation).

Reviewer #4 (Remarks to the Author):

I am satisfied with the revisions in response to my comments.

A minor suggestion: for the new Fig.5A-B, the number of peaks can be indicated, either in the legend or in the main text.

Reviewer #5 (Remarks to the Author):

The authors have done a thorough job of responding to reviewers' comments. They have taken comments to heart and made many helpful adjustments to the manuscript. Overall, the case supporting an important role for TRIM28 in maintenance of the ovarian pathway is very

strong.

In my view, the correlation with SUMOylation changes is the weakest aspect of the manuscript, and will remain so until the authors perform a genome-wide SUMOylation analysis in future work to assess the extent of these changes. Given the wide range of functions of TRIM28, and the fact that disruption of the PHD domain could affect other functions of the protein, I would favor adding a line in the discussion stating this and saying that a role of the protein beyond SUMOylation cannot be ruled out.

Point-by-point response to the reviewers.

We thank the referees for their reading of the revised version of the manuscript. As described in detail below, we addressed all their comments.

Reviewer #3 (Remarks to the Author):

In their revised manuscript and rebuttal, the authors have addressed my concerns by providing additional experiments, modifying their figures and the text. All the efforts provided are appreciated and improve the manuscript.

There is only one disagreement that should be revised in the text of the final manuscript version:

R3_4: "Have you looked at an earlier time-point than 20 days if SOX8 protein expression precedes the loss of FOXL2 protein rather than appearing after FOXL2 loss in supporting cells?" Response: In our immunofluorescent experiments, we never saw any cell expressing both FOXL2 and SOX8, as reported in the manuscript Rev#3-4.

I disagree with this response. Based on Fig. 1a, it is noticeable that at 20 dpp, many SOX8+ supporting cells are FOXL2 / SOX8 double positive, with a gradient of overlap transitioning from yellow/light green (high FOXL2, low SOX8) to dark orange (low FOXL2, high SOX8). See attached image for Fig. 1a with added arrows showing examples of yellow or orange double positive cells within the follicles.

This suggests that some supporting cells start expressing SOX8 before FOXL2 is fully turned off. Then by 8 w (Fig S3a), they are indeed mostly mutually exclusive as the authors mentioned in their rebuttal response. This observation provides input on the transdifferentiation event, which clearly is ongoing at 20 dpp, answering the question R2_12 from reviewer 2 that the Sertoli-like cells indeed originate from the differentiated granulosa cells within follicles, rather than from another source. This is also in agreement with other publications describing supporting cell transdifferentiation events in mouse gonads, for which a brief/transient co-expression of granulosa/Sertoli markers was observed. The presence of these double positive cells should be mentioned in the manuscript.

Other minor comment:

Fig. 5d: track information is too small to read (y axis, chromosomal position, gene orientation).

Response to reviewer #3

We are agreeing with reviewer #3 about FOXL2 / SOX8 double positive cell, it was a confusion that we made in the first revision. Thanks to him to underline this problem. Effectively in fig 1a some double stained cells for FOXL2 and SOX8 are clearly visible. In fact, we never seen cells that express both SOX9 and FOXL2. This not surprising as our results from figs 1 and 2 show that SOX9 is expressed after SOX8 in the transdifferentiation.

"The presence of these double positive cells should be mentioned in the manuscript."
-The following sentence has been added in the results of the manuscript (lane 4 page 5): "Some of these cells are co-expressing FOXL2 and SOX8, suggesting a transdifferentiation event. »
And in the legend of figure 1 we added "An overlap of both stainings is also visible, showing that some cells are co-expressing FOXL2 and SOX8."

“Fig. 5d: track information is too small to read (y axis, chromosomal position, gene orientation).”
-This has been corrected following recommendations of the editor.

Reviewer #4 (Remarks to the Author):

I am satisfied with the revisions in response to my comments.
A minor suggestion: for the new Fig.5A-B, the number of peaks can be indicated, either in the legend or in the main text.

Response to reviewer #4

We apologize that it was not clearly indicated in the legend of figure 5 that the number of peaks is already presented on the figure 5a and 5b. However, we think that it would be difficult to add it in the legend as it would be confusing for the reader.

Therefore, we added the following sentences on legend of figure 5a “The number of peaks analysed for each condition is reported on upper (blue or light red) of each chart”

For figure 5b we added: “The number of peaks analysed is the same as reported on figure 5a”

Reviewer #5 (Remarks to the Author):

The authors have done a thorough job of responding to reviewers' comments. They have taken comments to heart and made many helpful adjustments to the manuscript. Overall, the case supporting an important role for TRIM28 in maintenance of the ovarian pathway is very strong.

In my view, the correlation with SUMOylation changes is the weakest aspect of the manuscript, and will remain so until the authors perform a genome-wide SUMOylation analysis in future work to assess the extent of these changes. Given the wide range of functions of TRIM28, and the fact that disruption of the PHD domain could affect other functions of the protein, I would favor adding a line in the discussion stating this and saying that a role of the protein beyond SUMOylation cannot be ruled out.

Response to reviewer #5

“In my view, the correlation with SUMOylation changes is the weakest aspect of the manuscript, and will remain so until the authors perform a genome-wide SUMOylation analysis in future work to assess the extent of these changes”

We are not agreeing with this comment as it is what we have done in the last part of the manuscript (figure 5): we analysed genome wide the differential SUMOylation profile for SUMO1 and 2 between wild type, cKO and PHD mutant. We showed that down-regulated genes are preferentially Hypo-SUMOylated while up-regulated genes are preferentially Hyper-SUMOylated.

“Given the wide range of functions of TRIM28, and the fact that disruption of the PHD domain could affect other functions of the protein, I would favor adding a line in the discussion stating this and saying that a role of the protein beyond SUMOylation cannot be ruled out.”

We added the following sentence in the discussion (page 20 line 9-11) “So far, no function other than E3-SUMO ligase activity has been attributed to the PHD domain of TRIM28. We cannot exclude that this domain may have another function even if our genome wide analyses are strongly in favour of a major role of it E3 ligase activity”